# Towards Theoretical Understanding of Transformer Test-Time Computing: Investigation on In-Context Linear Regression

**Xingwu Chen** [* 1]  **Miao Lu** [* 2]  **Beining Wu** [3]  **Difan Zou** [4]

## Abstract

Scaling test-time computation during language model inference, such as generating intermediate thoughts or sampling multiple candidate answers, has proven effective in improving model performance. While these techniques inherently rely on the stochastic nature of inference to explore diverse reasoning paths, prior theoretical works typically build on a *deterministic decoding* framework, overlooking the stochastic nature of practical language model inference. This work takes an initial step to bridge this gap by establishing a new theoretical framework, incorporating *randomness and sampling* directly into the decoding analysis. To demonstrate the framework's effectiveness, we apply it to the canonical in-context linear regression task with continuous and binary coefficients, simulating decoding via noise injection and sampling to analyze widely adopted inference techniques. We validate our theoretical findings through numerical simulations, with additional experiments on real-world tasks substantiating the framework's potential for practical applications.

## 1. Introduction

Transformer-based (Vaswani, 2017) large language models (LLMs) have emerged as powerful general-purpose reasoning engines, achieving state-of-the-art performance across natural language processing (Singh et al., 2025; Guo et al., 2025; Yang et al., 2025a) and extending into domains such as computer vision (Peebles & Xie, 2023; Agarwal et al.,

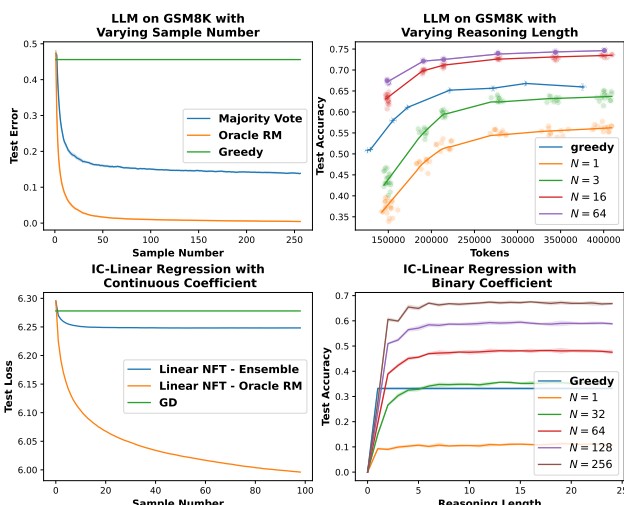

*Figure 1.* Comparison between real-world LLM's inference (above) and our designed sampling framework (below) for various test-time computing techniques with different sample numbers $N$ and reasoning lengths. Our framework exhibits trends similar to real-world LLMs' inference, Details can be found in Appendix C

2025). While scaling laws for LLM training (Kaplan et al., 2020) have traditionally characterized performance as a function of train-time compute, leveraging additional *test-time computation*, such as extending reasoning length via intermediate thoughts (Wei et al., 2022; Guo et al., 2025; OpenAI, 2024) or sampling and aggregating multiple candidate answers (Cobbe et al., 2021; Wang et al., 2023), has recently emerged as a new frontier. Empirical studies have formalized these gains into *inference scaling laws* (Wu et al., 2024; Yang et al., 2025b). However, despite this empirical success, the theoretical understanding of transformers, particularly regarding various *test-time computation* techniques, remains in its infancy.

The immense success of the transformer has catalyzed a line of theoretical work aiming to demystify its underlying mechanisms, ranging from analyzing expressive power (von Oswald et al., 2023; Liu et al., 2024; Li et al., 2025b) to investigating the training dynamics of transformers initialized from scratch (Yang et al., 2024; Zhang et al., 2023; Chen et al., 2024a; Huang et al., 2025). Most of these works focus

---

[*]Equal contribution  [1]School of Computing & Data Science, The University of Hong Kong [2]Department of Management Science and Engineering, Stanford University [3]Department of Statistics, University of Chicago [4]School of Computing & Data Science, Institute of Data Science, The University of Hong Kong. Correspondence to: Xingwu Chen <xingwu@connect.hku.hk>, Miao Lu <miaolu@stanford.edu>, Beining Wu <beiningw@uchicago.edu>, Difan Zou <dzou@cs.hku.hk>.

*Proceedings of the 43rd International Conference on Machine Learning*, Seoul, South Korea. PMLR 306, 2026. Copyright 2026 by the author(s).

on the in-context linear regression task (Garg et al., 2023) and build upon a theoretical framework assuming (simplified) transformer (von Oswald et al., 2023) with *deterministic* decoding procedure. While this framework successfully captures key operational mechanisms of attention (Han et al., 2024) and has proven effective to align and analyze practical LLMs (Dai et al., 2023; Yao et al., 2024b), it predominantly focuses on how transformers *directly* leverage output activations to solve specific tasks, thereby overlooking the stochastic nature of practical decoding procedures.

Crucially, such deterministic frameworks ignore the sampling and tokenization procedures inherent to language model decoding, creating a substantial gap between theoretical analysis and practical applications. Prior theoretical works typically assume a deterministic decoding procedure where the model output is fixed for a given prompt. In contrast, many practical inference techniques for scaling up test-time computing, such as majority voting (Wang et al., 2023), best-of-N sampling (BoN) (Cobbe et al., 2021), and tree of thoughts (ToT) (Yao et al., 2024a), rely fundamentally on probabilistic sampling. In these real-world scenarios, the model predicts subsequent tokens by computing a distribution over potential candidates and sampling from it. This discrepancy between deterministic theoretical setups and stochastic real-world LLM behavior hinders us towards understanding and analyzing of the success of transformer test-time computation.

**Our contributions.** In this work, we aim to bridge the gap between practical language model (probabilistic) inference and theoretical transformer analysis, providing initial theoretical insights into transformer test-time computation. We introduce a theoretical framework to formally study the effects of test-time computation by incorporating stochasticity into the transformer decoding process. To demonstrate the framework's effectiveness, we instantiate it on the canonical in-context linear regression task, modeling sampling-based decoding via noise injection and discrete sampling to analyze various inference techniques. Our main contributions are as follows:

- We establish a novel theoretical framework to bridge the gap between practical language model inference and theoretical transformer analysis. Our framework simulates language model decoding through noise injection and binary coefficient sampling, exhibiting trends similar to real-world LLMs' inference, as shown in Figure 1.
- We effectively apply this framework to analyze diverse inference techniques across theoretical settings. Through detailed case studies on in-context linear regression with both continuous and binary coefficients, we analyze how test-time computation scaling (i.e., increasing reasoning steps and sample counts) improves performance. Crucially, we provide theoretical proofs explaining when and

how sampling-based methods (Ensembling/ Majority Voting) outperform deterministic procedures (Constant reasoning and Greedy decoding), specifically by mitigating overfitting and escaping local optima.
- We validate our theoretical findings through extensive numerical experiments and demonstrate practical utility. Beyond validating our theorems, we leverage the theoretical insights from our framework to predict real-world LLM performance. The results demonstrate the potential of applying our theoretical framework for practical LLM behavior analysis.

**Related Works.** Our work is related to recent works on *scaling test-time computing in LLMs*, *theory for transformer test-time computing*, and *theory for transformer in-context learning*. Further details are provided in Appendix A.

## 2. Preliminaries

**In-context Linear Regression:** Drawing upon the capability of LLMs to learn from context with limited task demonstrations (Brown et al., 2020), in-context linear regression (Garg et al., 2023) considers a set of examples $\{(\mathbf{x}_i, y_i)\}_{i \in [n]}$:

$$(\mathbf{x}_i, y_i) \sim \mathbb{D}_{\mathbf{w}^*}, y_i = \mathbf{x}_i^\top \mathbf{w}^* + \epsilon_i, \ \forall i \in [n], \quad (2.1)$$

where $\mathbf{x}_i, \mathbf{w}^* \in \mathbb{R}^d$, $\mathbf{w}^*$ denotes the ground truth parameter sampled from a prior distribution $p_{\mathbf{w}}(\cdot)$, $\epsilon_i$ represents i.i.d. random noise, and $n \in \mathbb{N}$ is the number of in-context examples. The objective of the task is to either predict the label $y$ for a newly sampled $\mathbf{x}$ (Zhang et al., 2023; Chen et al., 2024b; Li et al., 2025b) or directly infer the ground truth $\mathbf{w}^*$ (Huang et al., 2025), given $n$ in-context pairs $(\mathbf{x}_i, y_i)$. In-context linear regression has served as a widely adopted task for theoretical transformer analysis, proving effective in capturing key transformer mechanisms (Han et al., 2024) and providing insights for practical LLM analysis (Dai et al., 2023; Yao et al., 2024b).

**Tansformer Architecture:** Transformers (Vaswani, 2017) are a type of neural network with stacked attention and multi-layer perceptron (MLP) blocks. In each layer, the transformer first utilizes attention module Attn to process the input sequence (or hidden states) $\mathbf{H} \in \mathbb{R}^{d \times t}$ with *sequence level* transformation:

$$\mathsf{Attn}(\mathbf{H}, \theta_1) = \mathbf{H} + \mathbf{V}\sigma(\mathbf{K}^\top \mathbf{Q}),$$

where $\mathbf{V} = \mathbf{W}_V \mathbf{H}, \mathbf{Q} = \mathbf{W}_Q \mathbf{H}, \mathbf{V} = \mathbf{V}_V \mathbf{H}, \theta_1 = \{\mathbf{W}_V, \mathbf{W}_K, \mathbf{W}_Q \in \mathbb{R}^{d \times d}\}$ are learnable parameters, and $\sigma(\cdot)$ denote the activation, such as Softmax, ReLU, or the identity function (linear attention). The MLP block subsequently applies a non-linear *token level* transformation:

$$\mathsf{MLP}(\mathbf{H}', \theta_2) = \mathbf{W}_1 \mathsf{ReLU}(\mathbf{W}_2 \mathbf{H}'),$$

where $\theta_2 = \{\mathbf{W}_1, \mathbf{W}_2\}$ denotes the parameters of MLP. For brevity, we omit layernorm and bias terms in this formulation.

**Tansformer Theoretical Simplifications** To ensure mathematical tractability, theoretical studies typically adopt the following architectural simplifications:

- **Attention-only Transformers**: as MLP layer represent token level operation and ignorant to the tokens in context, theoretical works typically removing this module and focusing on Attn module, resulting attention-only transformer (Von Oswald et al., 2023; Huang et al., 2023; 2025).
- **Simplified Attention Mechanism**: While vanilla Transformers utilize Softmax activation, theoretical analyses often simplify this to ReLU activation (Bai et al., 2024; Chen & Zou, 2024; Liu et al., 2024) or linear attention (identity map) (Von Oswald et al., 2023; Huang et al., 2025).

These simplifications are widely accepted in the theoretical community. Furthermore, empirical studies have demonstrated that these simplified models effectively validate practical transformer mechanisms and yield actionable insights for real-world applications (Ahn et al., 2023b; Yao et al., 2024b; Chen et al., 2024b).

**Practical Transformer Decoding Process** In practice, the inference process for a transformer can be abstracted into two main stages: computing the hidden states and sampling the next token via a sampling algorithm. Specifically, given the current input sequence embedding $\mathbf{H} = (\mathbf{h}_1, \ldots, \mathbf{h}_t) \in \mathbb{R}^{d \times t}$, the model *iteratively* performs the following two steps:

1. **Compute and extract the hidden state for the last position** $t$: Calculate $\widetilde{\mathbf{h}}_t = [\mathtt{TF}_\theta(\mathbf{H})]_t$, where $\mathtt{TF}_\theta$ denotes the transformer parameterized by $\theta$.
2. **Sample the next token** $x_{t+1}$ **(and thus its embedding** $\mathbf{h}_{t+1}$**):** This step samples based on the probability distribution returned by a sampling algorithm conditioned on $\widetilde{\mathbf{h}}_t$, denoted as $\mathbf{h}_{t+1} \leftarrow \mathtt{Sampling\_Alg}(\widetilde{\mathbf{h}}_t)$.

We emphasize that unlike most theoretical works (Von Oswald et al., 2023; Bai et al., 2024; Huang et al., 2025), which often assume the next token is directly determined by the hidden state $\widetilde{\mathbf{h}}_t$ (deterministic decoding), **practical decoding inherently involves a stochastic sampling algorithm** for auto-regressive next token prediction. This stochastic nature is fundamental to various techniques for scaling test-time compute to enhance Large Language Model (LLM) performance, such as Best-of-N (BoN) sampling (Stiennon et al., 2020; Nakano et al., 2021; Dong et al., 2023) and Majority Voting (Wang et al., 2022).

## 3. Theoretical Framework

In this section, we introduce our designed theoretical framework to analyze sampling-based test-time computing in transformers. Specifically, motivated by the recent work of (Huang et al., 2025), we consider the specific goal of *in-context coefficient prediction*, where the final output of the transformer reasoning path is a prediction $\widehat{\mathbf{w}}$ of the task coefficient $\mathbf{w}^*$. To handle the challenge, we explicitly construct an inference mechanism that involves both randomness and auto-regressive Chain-of-Thought (CoT) reasoning to solve in-context linear regression tasks. The transformer inference mechanism is designed to output stochastic reasoning paths, and different sampling-based test-time computing techniques correspond to how to aggregate different reasoning paths.

**Inputs and Task Setup.** Given an in-context dataset $\{(\mathbf{x}_i, y_i)\}_{i=1}^n$, we define the prompt embedding input to the transformer as the matrix $\mathbf{H}_0 \in \mathbb{R}^{d_e \times (n+1)}$, structured as follows:

$$\mathbf{H}_0 = \begin{pmatrix} \mathbf{x}_1 & \cdots & \mathbf{x}_n & \mathbf{0} \\ y_1 & \cdots & y_n & 0 \\ \mathbf{0} & \cdots & \mathbf{0} & \mathbf{w}_0 \\ 0 & \cdots & 0 & 1 \end{pmatrix} := \begin{pmatrix} \mathbf{X}^\top & \mathbf{0} \\ \mathbf{y}^\top & 0 \\ \mathbf{0} & \mathbf{w}_0 \\ \mathbf{0} & 1 \end{pmatrix}, \quad (3.1)$$

where the embedding dimension is $d_e = 2d + 2$. Here, $\mathbf{X}^\top = [\mathbf{x}_1, \ldots, \mathbf{x}_n] \in \mathbb{R}^{d \times n}$ denotes the matrix of covariates, and $\mathbf{y}^\top = [y_1, \ldots, y_n] \in \mathbb{R}^{1 \times n}$ represents the vector of corresponding labels. Without loss of generality, we initialize the coefficient estimate as $\mathbf{w}_0 = \mathbf{0}$.

The task objective is to recover the ground truth parameter $\mathbf{w}^*$ using a transformer $\mathtt{TF}_\theta$ with parameters $\theta$ (Huang et al., 2025). This process mimics CoT reasoning (Wei et al., 2022), where the intermediate estimates $\hat{\mathbf{w}}$ serve as reasoning steps. Specifically, for continuous coefficients $\mathbf{w}^* \in \mathbb{R}^d$, we aim to minimize the recovery loss $\frac{1}{2}\|\mathbf{w} - \mathbf{w}^*\|_{\mathbf{H}}^2$; for binary coefficients $\mathbf{w}^* \in \{0, 1\}^d$, we seek exact recovery (i.e., $\mathbf{w} = \mathbf{w}^*$). Here, $\mathbf{w}$ represents the coefficient predicted by $\mathtt{TF}_\theta$, extracted from the *decoded output embedding* at the last sequence position, denoted as $\mathbf{h}$. Specifically, $\mathbf{w}$ corresponds to the entries of $\mathbf{h}$ with indices from $d + 2$ to $2d + 1$ (i.e., $\mathbf{w} = \mathbf{h}_{d+2:2d+1}$).

**Stochastic Auto-Regressive Decoding Process.** Given the transformer model $\mathtt{TF}_\theta$ and the prompt $\mathbf{H}_0$ defined in (3.1), our framework simulates the decoding process of practical LLM via the following inference mechanism:

Here, $\mathtt{Sampling\_Alg}(\cdot)$ represents a specified sampling strategy that determines the distribution of the next token embedding, conditioned on the transformer's output hidden state at the final position.

---

**Algorithm 1** Auto-Regressive Decoding Framework

---

1: **Input:** Transformer $\mathtt{TF}_\theta$, Max steps $T$, Sampling Algorithm $\mathtt{Sampling\_Alg}(\cdot)$, Initial Prompt $\mathbf{H}_0$.
2: Initialize sequence: $\mathbf{H} \leftarrow \mathbf{H}_0$
3: **for** $t = 1$ to $T$ **do**
4:     Compute Hidden State: $\widetilde{\mathbf{H}} \leftarrow \mathtt{TF}_\theta(\mathbf{H})$
5:     Sample the next token embedding conditioned on the last hidden state: $\mathbf{h} \leftarrow \mathtt{Sampling\_Alg}(\widetilde{\mathbf{H}}_{:,-1})$
6:     Concatenate to obtain the embedding matrix for the new sequence: $\mathbf{H} \leftarrow (\mathbf{H}, \mathbf{h})$
7: **end for**
8: **Output:** $\mathbf{H}$.

---

We emphasize that **the output of this mechanism is determined jointly by the transformer model and the selected sampling algorithm**. Upon completing the iterative execution of Algorithm 1, the final coefficient prediction $\mathbf{w}$ is extracted from the decoded output embedding at the last position (i.e., $\mathbf{w} = \mathbf{H}_{d+2:2d+1,-1}$).

**Examples of Sampling-based Test-Time Computing Techniques within Our Framework.** Here we demonstrate how our framework can be applied to analyze standard test-time computing techniques, such as Ensemble or Majority Voting, through in-context linear regression.

Consider a transformer $\mathtt{TF}_\theta$, a sampling algorithm $\mathtt{Sampling\_Alg}(\cdot)$, and a prompt embedding matrix $\mathbf{H}_0$ as defined in (3.1). Given a reasoning step $t$ and a sampling budget $N \in \mathbb{N}_+$, the process begins by generating $N$ independent reasoning trajectories, $\{\mathbf{H}^{(j)}\}_{j=1}^N$, by executing Algorithm 1 $N$ times. Subsequently, the predicted coefficients, denoted as $\{\mathbf{w}_t^{(j)}\}_{j=1}^N$, are extracted from the decoded output embedding at the last sequence position of each trajectory.

Based on these generated samples $\{\mathbf{w}_t^{(j)}\}_{j=1}^N$, we formalize the test-time computing methods studied in this paper as distinct aggregation strategies:

- **Ensemble:** Aggregates predictions by computing their arithmetic mean:

$$\mathbf{w}_{\mathtt{avg}} := \frac{1}{N} \sum_{j=1}^N \mathbf{w}_t^{(j)}.$$

- **Majority Voting:** Selects the most frequently occurring prediction (the mode) from the set:

$$\mathbf{w}_{\mathtt{mv}} := \underset{\mathbf{w} \in \{\mathbf{w}_t^{(j)}\}_{j=1}^N}{\arg\max} \ \mathtt{Occur}(\mathbf{w}),$$

where $\mathtt{Occur}(\cdot) : \mathbb{R}^d \mapsto \mathbb{N}$ is a function counting the frequency of the input vector within the generated set.

The remainder of this paper utilizes this framework to analyze test-time computing for in-context linear regression.

In Section 4, we detail the specific sampling algorithms designed for this task. Subsequently, we present our theoretical analysis for continuous coefficients in Section 5 and for binary coefficients in Section 6.

## 4. Scaling Test-time Computation for In-Context Regression

Our theoretical framework can be effectively applied to analysis various inference techniques, in this section, we specific sampling algorithms for in-context linear regression with continous and binary coefficients, and investigate the effectiveness and the scaling law of the above test-time computing techniques.

### 4.1. Case Study 1: In-context Linear Regression with Continuous Coefficient

The first type of tasks we consider is the standard in-context linear regression with continuous regression coefficient sampled from a Gaussian distribution, i.e., $p_{\mathbf{w}} = \mathcal{N}(\mathbf{0}, \omega^2 \cdot \mathbf{I}_d)$. For this case, the specific type of sampling algorithms $\mathtt{Sampling\_Alg}$ we study is concluded in Algorithm 2.

---

**Algorithm 2** Sampling algorithm for in-context linear regression with continuous coefficient

---

1: **Input:** token embedding $\widetilde{\mathbf{h}}$, noise level $\sigma \geq 0$, noise transformation function $\phi.(\cdot) : \mathbb{R}^d \times \mathbb{R}^d \mapsto \mathbb{R}^d$.
2: Extract the coefficient $\widetilde{\mathbf{w}}$ from $\widetilde{\mathbf{h}}$, i.e., $\widetilde{\mathbf{w}} = (\widetilde{\mathbf{h}})_{d+2:2d+1}$
3: Sample a noise vector $\boldsymbol{\xi} \sim \mathcal{N}(\mathbf{0}, \sigma^2 \cdot \mathbf{I}_d)$
4: Define $\mathbf{w} \leftarrow \widetilde{\mathbf{w}} + \phi_{\boldsymbol{\xi}}(\widetilde{\mathbf{w}})$
5: **Output:** $\mathbf{h} := (\mathbf{0}, 0, \mathbf{w}, 1)^\top$.

---

Under the sampling framework of Algorithm 2, the next weight $\mathbf{w}$ is generated as $\mathbf{w} = \widetilde{\mathbf{w}} + \phi\boldsymbol{\xi}(\mathbf{w})$, where $\boldsymbol{\xi}$ is a Gaussian random seed and $\phi_{\boldsymbol{\xi}}$ is a noise transformation function (NFT). Here, the noise level $\sigma$ mimics the temperature used in practical language model sampling, where larger $\sigma$ introduces greater randomness into the reasoning path. The core intuition behind Algorithm 2 is that injecting noise into the auto-regressive decoding process enables exploration of the task's optimization landscape. We aim to investigate whether the test-time computing techniques can effectively aggregate these stochastic reasoning paths to achieve superior performance compared to a deterministic baseline by mitigating overfitting.

In this paper, we investigate the following two concrete and simple examples of the noise transformation function (NFT) $\phi_{\boldsymbol{\xi}}$. While potential future works could investigate other types of $\phi_{\boldsymbol{\xi}}$.

**Example 4.1** (Constant NFT). $\phi_{\boldsymbol{\xi}}(\mathbf{w}) := \boldsymbol{\xi}$, *independent of the input* $\mathbf{w}$ *and is homogeneous across reasoning steps.*

**Example 4.2** (Linear NFT). $\phi_{\boldsymbol{\xi}}(\mathbf{w}) := \boldsymbol{\xi}\boldsymbol{\xi}^\top \mathbf{w}$, *linear in the input predicted weight* $\mathbf{w}$ *such that the sampling distribution has different shape based upon the current decoding result.*

We consider the following test-time computing methods.

**Baseline: Direct Transformer Decoding.** This method employs the model's direct output as the next token by bypassing the sampling procedure in Algorithm 1 (i.e., skipping line 5 and directly utilizing the hidden state $\widetilde{\mathbf{H}}_{:,-1}$ as the next token $\mathbf{h}$). This deterministic procedure serves as a standard reference point without multiple sampling paths.

**Ensemble.** We consider sample average of the predictions from $N$ reasoning paths. We denote the resulting prediction after $N$ sampling paths of length $t$ as $\mathbf{w}_{\mathtt{avg}}$.

**Best-of-N.** We also consider BoN with the oracle reward model $R^\star(\mathbf{w}) := -\|\mathbf{w} - \mathbf{w}^*\|_2^2$. The resulting prediction accuracy gives an upper bound for other test-time computing method due to the usage of the truth. We denote the resulting prediction after $N$ sampling paths of length $t$ by $\mathbf{w}_{\mathtt{BoN}}$.

### 4.2. Case Study 2: In-context Sparse Linear Regression in Discrete Space

Motivated by the practical setting where the candidate tokens lie in a discrete space, we also consider another case in which the coefficient is a sparse binary vector, denoted as $\mathbf{w}^* \in \{0,1\}^d$ with $\|\mathbf{w}^*\|_0 = k < d$. In this situation, we consider the following sampling algorithm $\mathtt{Sampling\_Alg}$, which performs sampling on a discrete space $\{0,1\}^d$ based on the predicted weight $\widetilde{\mathbf{w}}$ in the transformer output.

---

**Algorithm 3** Sampling algorithm for in-context linear regression with binary coefficient

---

1: **Input:** token embedding $\widetilde{\mathbf{h}}$, coefficient sparsity $k \in [d]$.
2: Initialize $\mathbf{w} \leftarrow \mathbf{0}_d$
3: Extract the coefficient $\widetilde{\mathbf{w}}$ from $\widetilde{\mathbf{h}}$, i.e., $\widetilde{\mathbf{w}} = (\widetilde{\mathbf{h}})_{d+2:2d+1}$
4: Compute predicted distribution $p = \mathtt{ClipNorm}(\widetilde{\mathbf{w}})$
5: Sample $k$ different indices $(e_1, \ldots, e_k) \subset [d]$ based on $p$ without replacement
6: Assign $w_{e_\ell} = 1$ for each $e_\ell \in \{e_1, \cdots, e_k\}$
7: **Output:** $\mathbf{h} := (\mathbf{0}, 0, \mathbf{w}, 1)^\top$.

---

In algorithm 3, the function $\mathtt{ClipNorm}(\cdot)$ first clips each element in $\widetilde{\mathbf{w}}$ to be non-negative and then normalizes the resulting vector such that its elements sum to 1, i.e., $(\mathtt{ClipNorm}(\widetilde{\mathbf{w}}))_i = \max\{\widetilde{w}_i, 0\} / \sum_{i'=1}^d \max\{\widetilde{w}_{i'}, 0\}$. This resembles the softmax operation over a vocabulary set. Then algorithm 3 simulates LLM decoding by sampling tokens based such a distribution. More specifically, given the distribution $p$, we sample the (embedded) next token $\mathbf{w}$ as a $k$-sparse vector with non-zero coordinates sampled from $p$. We treat the vector sparsity $k$ as a fixed parameter satisfying

$1 \le k < d$, with $k$ typically set to 1 in practice. Such a discrete nature of these coefficients enables us to consider more methods such as majority voting among the sampling-based test-time computing strategies. In this work, we compare majority voting to a baseline inference mechanism based on greedy decoding which does not utilize sampling.

**Baseline: greedy decoding.** In the decoding step, instead of sampling $k$ items based on $p$ as depicted in Algorithm 3 (Line 3), we opt to choose $k$ items with the highest $k$ probabilities under $p$ and set the corresponding indices of $\mathbf{w}$ to 1. This mirrors the greedy decoding algorithm commonly used in practice. We denote the resulting prediction after $t$ reasoning steps as $\mathbf{w}_t^{\mathtt{greedy}}$.

**Majority voting.** Utilizing the discrete nature of the coefficients, we apply the $\mathtt{Occur}(\cdot)$ function to candidate answers, selecting the most frequent one as our majority voting. The prediction after sampling $N$ reasoning paths of length $t$ is denoted as $\mathbf{w}_{t,N}^{\mathtt{mv}}$.

### 4.3. Analysis through the Lens of a Simplified Transformer

Since practical test-time computing techniques typically build upon well-trained LLMs(Wang et al., 2022; Wei et al., 2022; Snell et al., 2024b), we initiate our theoretical analysis using a one-layer linear attention transformer (Von Oswald et al., 2023; Huang et al., 2025) as a representative model. Defined in Equation 4.1, we highlight that our framework can extend to more complex transformers, which could be the potential direction for our further work.

The model we consider is a one-layer self-attention module equipped with residual connection :

$$\mathtt{TF}_\theta(\mathbf{H}) := \mathbf{H} + \mathbf{V}\mathbf{H} \cdot \frac{\mathbf{H}^\top \mathbf{W}\mathbf{H}}{n} : \mathbb{R}^{d_e \times *} \mapsto \mathbb{R}^{d_e \times *}. (4.1)$$

Here, $\theta = \{\mathbf{V}, \mathbf{W}\}$ denotes the model parameters. Specifically, $\mathbf{V}, \mathbf{W} \in \mathbb{R}^{d_e \times d_e}$ represent the consolidated projection-value and key-query matrices, respectively. Towards the goal of in-context weight prediction (2.1), we introduce the following proposition, demonstrating that a well-trained one-layer self-attention Transformer inherently implements Chain-of-Thought style gradient descent.

**Proposition 4.3** (Well-Trained Transformer (4.1) Can Implement Gradient Descent ). *There exists a well-trained transformer instance* $\mathtt{TF}_{\theta_{\mathtt{GD}}}$ *in form of* (4.1). *such that given prompt* $\mathbf{H}_0$ *defined in* (3.1)*, the output embedding* $\widetilde{\mathbf{H}}_t \leftarrow \mathtt{TF}_\theta(\mathbf{H}_{t-1})$ *generated after* $t$ *iterative steps according to Algorithm 1 yields raw predicted weight* $\widetilde{\mathbf{w}}_t = (\widetilde{\mathbf{H}}_t)_{d+2:2d+1,-1}$ *that satisfies:*

$$\widetilde{\mathbf{w}}_t = \mathbf{w}_{t-1} - \frac{\eta}{n} \cdot \mathbf{X}^\top (\mathbf{X}\mathbf{w}_{t-1} - \mathbf{y})$$

*where the $\mathbf{w}_{t-1} = (\theta(\mathbf{H}_{t-1})_{d+2:2d+1,-1}$ denotes the predicted weight extracted from the decoded sequence embedding $\mathbf{H}_{t-1}$.*

This proposition can be directly validate by Theorem 3.2 and Theorem 4.1 from (Huang et al., 2025). We provide a comprehensive proof of Proposition 4.3 in Appendix D. Such gradient-based optimization serves as a fundamental mechanism implemented by both linear attention transformers (von Oswald et al., 2023) and practical LLMs (Dai et al., 2023). Consequently, $\mathtt{TF}_{\theta_{\mathtt{GD}}}$ serves as a robust foundational model for investigating Transformer test-time computation behaviors. While we utilize such simplified architecture, our framework is inherently extensible to more complex Transformer variants, representing a promising avenue for future research.

Next, we present theoretical results by taking $\mathtt{TF}_{\theta_{\mathtt{GD}}}$ as well-trained transformer start point for Case Study 1 and 2 in Section 5 and 6 respectively, with numerical experiments in Section 7.

## 5. Analysis of In-context Linear Regression with Continuous Coefficient

In this section, we establish the theoretical analysis for Section 4.1. We measure the performance of any in-context coefficient prediction by its population risk under $\mathbb{D}_{\mathbf{w}^*}$, i.e., $L_{\mathbb{D}_{\mathbf{w}^*}}(\mathbf{w}) := (1/2) \cdot \mathbb{E}_{(\mathbf{x},y)\sim\mathbb{D}_{\mathbf{w}^*}}[(y - \mathbf{x}^\top \mathbf{w})^2]$, which is equivalent to consider the following excess risk,

$$\mathcal{E}(\mathbf{w}) := L_{\mathbb{D}_{\mathbf{w}^*}}(\mathbf{w}) - \inf_{\mathbf{w}' \in \mathbb{R}^d} L_{\mathbb{D}_{\mathbf{w}^*}}(\mathbf{w}) = \frac{1}{2} \cdot \|\mathbf{w} - \mathbf{w}^*\|_{\mathbf{H}}^2,$$

where $\mathbf{H} := \mathbb{E}_{\mathbf{x}\sim\mathbb{D}_{\mathbf{w}^*}}[\mathbf{x}\mathbf{x}^\top]$ denotes the population covariance matrix. We denote the collection of label noise in the in-context data as $\boldsymbol{\epsilon} := \mathbf{y} - \mathbf{X}\mathbf{w}^*$. We also denote the eigenvalues of the population covariance matrix $\mathbf{H}$ as $\{\lambda_i\}_{1\leq i\leq d}$ in a non-increasing order. Our analysis relies on standard assumptions on the data distribution (Bartlett et al., 2020), which is presented in Assumption F.1 due to space limit. By the same reason, we present our results for a special case of $\mathbf{H}$ with polynomially decaying eigenvalues, and refer to the readers to the expressions of general $\mathbf{H}$ in Appendix E.

**Baseline: multi-step GD with CoT.** The following result gives the excess risk bound for transformers implementing vanilla multi-step gradient descent (D.1). This is a corollary of Theorem E.1 and is proved in Appendix F.2.

**Proposition 5.1.** *Under the same assumptions and setups as in Theorem E.1, by additionally assuming that the spectrum of $\mathbf{H}$ satisfies polynomially decaying, i.e., $\lambda_i = i^{-(r+1)}$ for some $r \geq 1$, then for any reasoning path length $t \lesssim \eta(r+$*

$1)^{(r+1)/2}d^{(r+1)/2}$, with probability at least $1 - 1/\mathrm{poly}(n)$,

$$\mathbb{E}_{\boldsymbol{\epsilon},\mathbf{w}^*}[\mathcal{E}(\mathbf{w}_{\mathtt{GD}})] \lesssim \omega^2 \cdot \left(\frac{1}{t\eta}\right)^{\frac{r}{r+1}} + \frac{\sigma_\epsilon^2}{n} \cdot (t\eta)^{\frac{1}{r+1}}.$$

**Aggregating by ensembling.** In this case, the final regression coefficient reasoned by the transformer test-time computing under the budget of CoT length $t$ and reasoning path number $N$ is explicitly given by $\mathbf{w}_{\mathtt{avg}} := N^{-1} \cdot \sum_{j=1}^N \mathbf{w}_t^{(j)}$, where each random reasoning path $\{\mathbf{w}_\ell^{(j)}\}_{1\leq\ell\leq t}$ is i.i.d. generated by executing Algorithm 1 on a well trained transformer satisfying Proposition 4.3 and with Algorithm 2. The following result gives the excess risk bound for this method with different choices of the NFT $\phi_{\boldsymbol{\xi}}$. The proof is in Appendix F.4.

**Theorem 5.2.** *Under the same assumptions and setups as in Theorem E.2, additionally assuming that the spectrum of $\mathbf{H}$ satisfies polynomially decaying, i.e., $\lambda_i = i^{-(r+1)}$ for some constant $r \geq 0$, we have the following results.*

1. *Constant noise transformation function (Example 4.1): taking the reasoning length $t \lesssim \eta(r+1)^{(r+2)/2}n^{(r+1)/2}$, with probability at least $1 - 1/\mathrm{poly}(n)$,*

$$\mathbb{E}\left[\mathcal{E}(\mathbf{w}_{\mathtt{avg}})\right] \lesssim \omega^2 \cdot \left(\frac{1}{t\eta}\right)^{\frac{r}{r+1}} + \frac{\sigma_\epsilon^2}{n} \cdot (t\eta)^{\frac{1}{r+1}} + \frac{\vartheta_{n,t}}{N}.$$

2. *Linear noise transformation function (Example 4.2): taking the noise variance $\sigma^2 \asymp d^{-1}$, the reasoning length $t > \sigma^{-2} \cdot \log 2$, with probability at least $1 - 1/\mathrm{poly}(n)$,*

$$\mathbb{E}[\mathcal{E}(\mathbf{w}_{\mathtt{avg}})] \lesssim \omega^2 \cdot \widetilde{\lambda}^{\frac{r}{r+1}} + \frac{\sigma_\epsilon^2}{n} \cdot \left(\frac{\eta(1-\sigma^2)}{\sigma^2}\right)^{\frac{1}{r+1}} + \frac{\varsigma_n}{N},$$

*where $\widetilde{\lambda} := \eta^{-1}(2t^{-1} + \sigma^2(1 + 2t^{-1})/(1 - \sigma^2))$.*

*Here the expectation is taken with respect to $\boldsymbol{\epsilon}$, $\mathbf{w}^*$, and all the sampling noise $\boldsymbol{\xi}$ across different reasoning steps and paths. The explicit formula for the functions $\vartheta_{n,t}$ and $\varsigma_n$ are deferred to (E.2) and (E.3), respectively.*

The above theorem reveals how the prediction accuracy evolves as the reasoning length $t$ and sample numbers $N$ increase. In particular, we make the following remarks (i) In the above excess risk, the terms $\vartheta_{n,t}/N$ and $\varsigma_n/N$ represent the error from sampling finitely many reasoning paths $N$. By taking $N$ large enough (see (E.4) and (E.5) in Corollary E.3), the leading term of the excess risk would be the first two terms. (ii) By the result for Example 4.1, Algorithm 2 with constant noise does not provide benefit compared with TF implementing vanilla GD (see Proposition 5.1). (iii) In contrast, we next show that with linear

NFT Algorithm 2 can prevent overfitting to noisy labels. Considering the following regime of the parameters,

$$\omega, \sigma_\epsilon \asymp 1, \quad n \asymp \eta d, \quad \sigma^2 \asymp d^{-1}, \quad t \asymp \widetilde{t} \cdot \sigma^{-2}, \text{(5.1)}$$

risk bounds for the vanilla multi-step GD and the ensemble method (using linear NFT (Example 4.2)) are as following,

$$\mathbb{E}_{\epsilon, \mathbf{w}^*} \left[ \mathcal{E}(\mathbf{w}_{\mathtt{GD}}) \right] \lesssim \widetilde{t}^{\frac{1}{r+1}} \cdot (\eta d)^{-\frac{r}{r+1}},$$

$$\mathbb{E}_{\epsilon, \mathbf{w}^*, \boldsymbol{\xi}} \left[ \mathcal{E}(\mathbf{w}_{\mathtt{avg}}) \right] \lesssim (\eta d)^{-\frac{r}{r+1}}, \text{ if } N \geq \eta^{\frac{r}{r+1}} d^{\frac{2r+1}{r+1}}.$$

Notice that by the conditions in Proposition 5.1 and Theorem 5.2, all the above conclusions hold when $t = \widetilde{t} \cdot \sigma^{-2}$ is not exceeding the order of $\eta (r+1)^{(r+1)/2} n^{(r+1)/2}$, which, under the parameter regime (5.1), translates to $\widetilde{t} \lesssim d^{(r-1)/2}$. Consequently, we observe that in high-dimensional settings, **the deterministic baseline (GD) suffers from harmful overfitting to label noise as the effective reasoning length $\widetilde{t}$ increases, whereas sampling-based test-time computation effectively mitigates overfitting to sustain performance** (see Remark E.4 for further details).

# 6. Analysis of In-context Sparse Linear Regression in Discrete Space

In this section, we conduct a theoretical analysis for binary sparse in-context linear regression (Section 4.2). Our strategy of studying and comparing the test-time computing methods is to analyze the probability of perfectly recovering the true coefficient, i.e., $\mathbb{P}(\mathbf{w}_t^{\mathtt{greedy}} = \mathbf{w}^*)$ and $\mathbb{P}(\mathbf{w}_{t,N}^{\mathtt{mv}} = \mathbf{w}^*)$. We use the notation $p(\mathbf{w}_t = \mathbf{w}) := \mathbb{P}(\mathbf{w}_t = \mathbf{w} \mid \mathbf{w}_0, \mathcal{D})$ to indicate the probability of weight $\mathbf{w}$ after $t$ reasoning steps, conditioning on the initial state $\mathbf{w}_0$ and the in-context dataset $\mathcal{D}$ in a single reasoning path. We define $\mathcal{W} = \{\mathbf{w} \mid \mathbf{w} \in \{0,1\}^d, \|\mathbf{w}\|_0 = k\}$ and assume $\boldsymbol{x} \sim \mathcal{N}(0, \mathbf{I}_d)$ and label noise $\epsilon_i \sim \mathcal{N}(0, \sigma_\epsilon^2)$ with $\sigma_\epsilon > 0$.

Our first result shows that if in a single reasoning path the prediction $\mathbf{w}_t$ has a probability of recovering the truth higher than that of recovering any other coefficient, then majority vote recovers the truth with a probability converging to 1 exponentially fast. The proof is in Appendix G.1.

**Proposition 6.1** (Sample complexity for majority vote)**.** *Consider the binary sparse in-context linear regression task (Section 4.2) and using majority vote with reasoning length $T$ and sampling number $N$. The final prediction $\mathbf{w}_{t,N}^{\mathtt{mv}}$ can asymptotically recover the truth $\mathbf{w}^*$ with probability 1 given sufficient sample size $N$ if for a single reasoning path*

$$\Delta_t := p(\mathbf{w}_t = \mathbf{w}^*) - \max_{\mathbf{w}' \in \mathcal{W} \setminus \{\mathbf{w}^*\}} p(\mathbf{w}_t = \mathbf{w}') > 0. \text{(6.1)}$$

*Under condition (6.1), it holds that*

$$\mathbb{P}\left(\mathbf{w}_{t,N}^{\mathtt{mv}} = \mathbf{w}^* \mid \mathbf{w}_0, \mathcal{D}\right) \geq 1 - |\mathcal{W}| \cdot \exp\left(-N\Delta_t^2/2\right).$$

We remark that similar results of Proposition 6.1 have also been proposed in (Wu et al., 2024). Here, we further provide more detailed analysis for the majority vote in our binary sparse linear regression task, show its dependence on the in-context example number $n$, reasoning length $t$, and compare it with the greedy decoding algorithm to emphasize when it is important to use the sample-then-select method.

Our main result to this end is the following two theorems. The first result is regarding the regime where we have sufficiently many in-context data $n$, with proof in Appendix G.2.

**Theorem 6.2** (Perfect recovery probability with sufficient in-context examples)**.** *Suppose that $n \geq (6k + 3\sigma_\epsilon)^4$, then the overall recovery probability of greedy decoding and majority vote are lower bounded as following:*

- ***Greedy decoding:** for any reasoning length $t \geq 1$, $\mathbb{P}(\mathbf{w}_t^{\mathtt{greedy}} = \mathbf{w}^*) \geq 1 - \delta(n)$*
- ***Majority vote:** for any reasoning length $t \geq 1$ and sampling number $N \geq 1$, it holds that*

$$\mathbb{P}\left(\mathbf{w}_{t,N}^{\mathtt{mv}} = \mathbf{w}^*\right) \geq \left(1 - \delta(n)\right) \cdot \left(1 - |\mathcal{W}| \cdot e^{-N\Delta_t^2/2}\right).$$

*Here $\delta(n) = 2d(d+2) \cdot \exp(-c \cdot n^{1/2})$ for some absolute constant $c > 0$, and for any $t \geq 1$, $\Delta_t$ satisfies that*

$$\Delta_t \geq \frac{p_{\mathtt{trans}}}{p_{\mathtt{trans}} + 1 - p_{\mathtt{recurr}}} \left(1 - (p_{\mathtt{recurr}} - p_{\mathtt{trans}})^{t-1}\right),$$

*where the quantities $p_{\mathtt{trans}}, p_{\mathtt{recurr}} \in (0,1)$ are defined as*

$$p_{\mathtt{trans}} := \left(1 - \frac{2k + \sigma_\epsilon}{n^{1/4} - (2k + \sigma_\epsilon)}\right) \cdot \frac{1}{d^k},$$

$$p_{\mathtt{recurr}} := \left(1 - \frac{\sigma_\epsilon}{n^{1/4} - \sigma_\epsilon}\right) \cdot \left(\frac{n^{1/4} - \sigma_\epsilon}{n^{1/4} - \sigma_\epsilon + d\sigma_\epsilon}\right)^k.$$

Theorem 6.2 establishes lower bounds on the recovery probability for both greedy decoding and majority vote. The recovery probability improves exponentially with the number of in-context examples. For majority vote, since $0 < \Delta_t < 1$ for all $t \geq 1$, as with sufficiently many number of sampling paths ($N \to \infty$), we have $\mathbb{P}\left(\mathbf{w}_{t,\infty}^{\mathtt{mv}} = \mathbf{w}^*\right) \geq 1 - \delta$, which matches that of greedy decoding $\mathbb{P}(\mathbf{w}_t^{\mathtt{greedy}} = \mathbf{w}^*)$, and both algorithms can achieve perfect accuracy given sufficient in-context examples $n$. Moreover, we remark that $p_{\mathtt{recurr}} > p_{\mathtt{trans}}$ since it holds that $(n^{1/4} - \sigma_\epsilon)(n^{1/4} - \sigma_\epsilon + d\sigma_\epsilon)^{-1} > d^{-1}$ for sufficiently many in-context examples $n > (3\sigma_\epsilon)^4$. When $\sigma_\epsilon = 0$, we have $p_{\mathtt{recurr}} = 1$ and $p_{\mathtt{trans}} > 1/2d^k$, ensuring that $\Delta_t$ converges to 1 as $t \to \infty$.

The theorem for sufficient in-context data does not highlight the advantage of majority vote in terms of recovery probability. However, real-world applications and our experiments show that majority vote is more accurate and robust with limited in-context data. We present our second main theorem to analyze this scenario, considering the case with only

one in-context example ($n = 1$ and $k = 1$). Although simplified, this case offers valuable insights into the robustness of majority vote.

**Theorem 6.3** (Majority vote outperforms greedy decoding in the case of limited in-context examples). *Consider the case where $n = k = 1, \sigma_\epsilon = 0$, and denote the in-context example as $(\mathbf{x}, \mathbf{x}^\top \mathbf{w}^*)$. We have the following results.*

- *Greedy decoding: for any reasoning length $t \geq 1$,*

$$\mathbb{P}\big(\mathbf{w}_t^{\mathtt{greedy}} = \mathbf{w}^*\big) \leq \frac{1}{2^{d-1}} + \frac{2}{d}.$$

- *Majority vote: there exists a $\zeta > 0$ such that for reasoning steps $t \geq 2 \log 2 / \log(1 - \zeta)$, sampling number $N \geq 1$,*

$$\mathbb{P}\big(\mathbf{w}_{t,N}^{\mathtt{mv}} = \mathbf{w}^*\big) \geq 1 - \frac{1}{2^{d-1}}.$$

Theorem 6.3 is proved in detail in Appendix G.3. The core intuition is that **deterministic greedy decoding often becomes trapped in cyclic state transitions**, preventing it from exploring a broader space to reach the optimal state $\mathbf{w}^*$. In the high-probability event constructed in Appendix G.3, greedy decoding repeatedly selects between two incorrect states $\mathbf{w}'$ and $\mathbf{w}''$; therefore, increasing the reasoning length alone cannot recover $\mathbf{w}^*$ once this cycle occurs. In contrast, the **stochasticity inherent in majority voting facilitates a more effective exploration of the state space**, ensuring high-probability convergence to $\mathbf{w}^*$ even in constrained, data-limited scenarios. These insights are further substantiated by numerical simulations in Section 7.1.

# 7. Experiments

## 7.1. Numerical Results for In-Context Linear Regression

Here, we validate our theoretical findings through numerical experiments. For the continuous case, we examine the effects of varying $\sigma_\epsilon$ and $\sigma$. Our results demonstrate that with ensemble aggregation, constant NFT provides no performance improvement, while linear NFT reduces test loss given sufficient sample size, confirming Corollary 5.2. Furthermore, when decoding with a reward model, even constant NFT yields consistent performance improvements as sample numbers increase.

For the binary sparse coefficient case, we observe from Fig 2 (e) that with sufficient examples, both greedy decoding and majority voting achieve perfect accuracy, supporting Theorem 6.2. From Fig 2 (f) we find that when setting $n = 1$ and $d = 10, \sigma_\epsilon = 0$, with sufficiently large reasoning length $T$, majority voting achieves high accuracy, while greedy search maintains approximately $2/d = 0.2$ accuracy, consistent with Theorem 6.3. We fit the relationship between accuracy Acc and sample number $N$ using $\mathtt{Acc} = \alpha_T - \beta_T e^{-\nu_T N}$ for given $T$. The results, shown in Fig 2 (g) and

(h), not only validate Theorem 6.1 but also suggest practical applications for real-world LLM inference.

## 7.2. Insights for LLM Inference

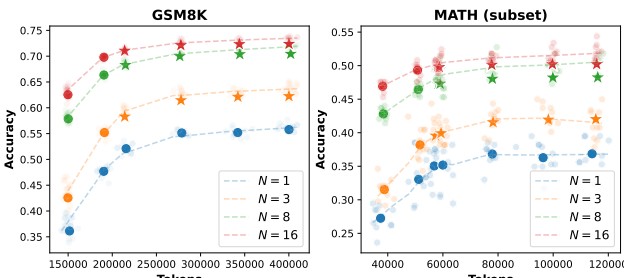

*Figure 3.* Utilizing data with low computational costs to forecast results for high computational costs, where ★ denotes predicted results and ● denotes the data utilized.

Our theoretical analysis identifies two critical terms governing the model's behavior: $\mathcal{O}(e^{-\Delta_T^2 N/2})$ and $\mathcal{O}(e^{-\mu T})$, which determine the overall accuracy $\mathtt{Acc}(T, N)$ and probability gap $\Delta_T$. Motivated by these bounds, we model the dependence on reasoning length as

$$\Delta_T \approx \gamma - \kappa e^{-\mu T},$$

and then model the accuracy under sampling budget $N$ as

$$\mathtt{Acc}(T, N) \approx \alpha - \beta e^{-\Delta_T^2 N/2}.$$

These equations form the mathematical backbone of our Low-Cost-to-High Prediction Algorithm (Algorithm 4; detailed in Appendix C.2): low-cost empirical observations with small $T$ or $N$ are used to fit the free parameters, and the fitted model extrapolates performance under high-compute settings. Compared with existing inference scaling laws that primarily model the effect of increasing the sample count $N$ (Wu et al., 2024), our formulation jointly captures the interaction between reasoning length $T$ and sampling number $N$. As illustrated in Fig. 3, the resulting predictions align well with real-world LLM performance.

# 8. Conclusions and Limitations

This paper takes an initial step toward bridging the gap between practical language model test-time computation and theoretical Transformer analysis by introducing a framework that incorporates randomness into the decoding process. Our framework effectively analyzes various inference techniques in different theoretical settings. We use it to conduct a detailed analysis of how test-time computation, including reasoning steps and sampling number, plays a role in Transformer reasoning. This analysis offers new insights

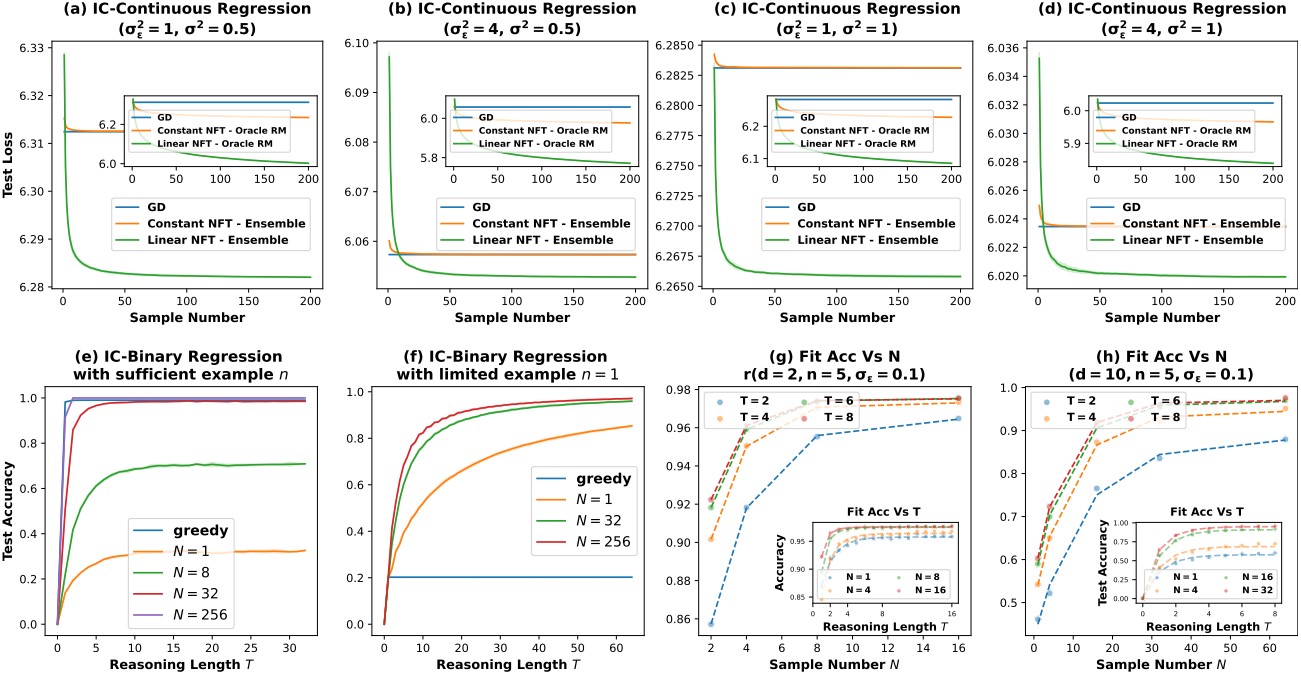

*Figure 2.* Numerical experiments on in-context linear regression with continuous coefficients (*a-d*) and binary coefficients (*e-h*).

into the inference behavior of real-world language models. Potential future works include analyzing other types of sampling algorithms and reasoning methods. Also it remains open to rigorously analyze the benefits of BoN method and its variants (with respect to different reward models) that we experimentally verified to be effective.

## Impact Statement

This paper presents work whose goal is to advance the field of Machine Learning. In particular, we aim to initiate the theoretical study of inference-time computation in LLMs, which can be leveraged to gain a deeper understanding of LLM inference and to guide or evaluate new reasoning methods. A potential negative impact of scaling test-time computation is that larger sampling budgets and longer reasoning traces can increase inference cost, energy consumption, and carbon footprint. These costs may also create accessibility disparities between well-resourced users who can afford extensive inference-time computation and users operating under constrained budgets. Our Low-Cost-to-High Prediction method may help mitigate these concerns by estimating high-compute performance from low-compute observations, enabling practitioners to choose more compute-efficient inference settings.

## Acknowledgments

We would like to thank the anonymous reviewers and area chairs for their helpful comments. We acknowledge the support from NSFC 62306252, Hong Kong ECS award 27309624, Guangdong NSF 2024A1515012444, and the central fund from HKU.

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

# A. Related Works

**Scaling test-time computing in LLMs.** Scaling test-time computing has demonstrated tremendous empirical success in LLMs, especially for reasoning tasks (OpenAI, 2024; Guo et al., 2025). Recent research on increasing test-time computing in LLMs primarily focuses on two aspects (Snell et al., 2024a): (i) generating longer reasoning paths, including chain-of-thought (CoT) prompting (Wei et al., 2022; Kojima et al., 2022) and self-refinement methods (Madaan et al., 2023; Saunders et al., 2022; Kumar et al., 2024); and (ii) generating multiple potential reasoning paths and selecting the optimal one through consistency-based selection (Wang et al., 2023), reward-guided choosing (Stiennon et al., 2020; Liu et al., 2020; Cobbe et al., 2021; Dong et al., 2023), or reasoning tree search (Yao et al., 2024a; Zhou et al., 2023). Empirical studies establish that increased test-time computation consistently improves model performance (Snell et al., 2024b; Yue et al., 2024; OpenAI, 2024), obeying formalized *inference scaling laws* (Wu et al., 2024; Yang et al., 2025b). Furthermore, recent work by Singhi et al. (2025) derives scaling laws for generative verification, suggesting that for compute-optimal inference, scaling the number of candidate solutions is often more effective than scaling verification depth. This paradigm shift highlights the necessity of theoretical frameworks that can model the trade-offs between generation diversity and verification, a gap our work aims to fill.

**Theory for transformer test-time computing.** Inspired by the empirical success of the inference-time computing techniques of LLMs, recently there have been a few works trying to demystify the mechanism behind it through analysis on theoretical tasks and simple transformer models. Wen et al. (2024) and Kim & Suzuki (2024) analyze how one-layer transformers utilize CoT to efficiently solve the $k$-parity task. More recently, (Huang et al., 2025) proved that transformers trained via gradient descent can generalize CoT to lengths unseen during training via recursive self-improvement, provided the task possesses specific algebraic structures. Mechanistically, (Zhang et al., 2025b) identified that transformers implementing CoT effectively learn internal Finite State Automata (FSAs), with late-layer MLPs robustly tracking the problem state. Hu et al. (2024) and Liu et al. (2025) study the statistical properties of CoT and its variants like majority vote, with Rautenberg & Schippkus (2026) providing rigorous probabilistic guarantees for the exponential error reduction achieved by independent prompt repetition. Our work differs by proposing a unified framework that incorporates randomness directly into the inference stage of transformer models, allowing us to study sophisticated test-time computing methods involving random sampling and ensembling within a rigorous linear regression setting.

**Theory for in-context learning by transformers.** In-context learning (ICL) (Brown et al., 2020) is a key capability of LLMs which means that the model is able to answer a new query provided with a few query-answer demonstrations of the similar tasks without updating the model parameters. The empirical success of ICL has sparked extensive theoretical research, much of which builds on the framework of (Garg et al., 2023), where transformers learn an unknown function $f(\cdot)$ from input-output pairs $\{(\mathbf{x}_i, f(\mathbf{x}_i))\}_{i=1}^n$. This formalism has enabled rigorous analysis across multiple dimensions, including expressive power (Bai et al., 2024; Guo et al., 2023), training dynamics (Zhang et al., 2023; 2025a; Huang et al., 2023; Chen et al., 2024a; Wu et al., 2023; Zhang et al., 2025a; 2026; Zhang & Cao, 2026), and mechanistic interpretations (Giannou et al., 2024; Li et al., 2025b). While early mechanistic work established an equivalence between linear self-attention and gradient descent (GD) (von Oswald et al., 2023; Ahn et al., 2023a), recent scholarship has significantly broadened this understanding. (Dragutinović et al., 2025) extended this equivalence to classification with softmax attention, demonstrating that transformers implement context-adaptive *kernel* gradient descent, while (Li et al., 2025a) provided a universal approximation theory for ICL, framing it as an in-context Lasso solver. Crucially, the field is moving beyond deterministic optimization toward probabilistic interpretations: (Reuter et al., 2025) showed that transformers can perform full Bayesian inference in-context to approximate complex posterior distributions, and (Kiruluta, 2026) formulated ICL as online Bayesian state estimation via Kalman filtering. These works suggest that deterministic GD is a degenerate case of a broader probabilistic inference mechanism. Our framework leverages this insight by modeling the "noise" in reasoning steps not as error, but as necessary exploration of the posterior distribution, bridging the gap between Bayesian ICL theory and stochastic decoding practices.

# B. Notation Table

*Table 1.* Summary of frequently used notation.

| Notation | Meaning |
|---|---|
| $n$ | Number of in-context examples. |
| $d$ | Dimension of the regression coefficient. |
| $k$ | Sparsity level of the binary coefficient. |
| $T$ or $t$ | Reasoning length or number of auto-regressive decoding steps. |
| $N$ | Number of sampled reasoning paths. |
| $\sigma$ | Noise level used in the continuous noise transformation function. |
| $\sigma_\epsilon$ | Standard deviation of the label noise. |
| $\mathbf{w}^*$ | Ground-truth regression coefficient. |
| $\mathbf{w}_t$ | Coefficient after $t$ reasoning steps in a single path. |
| $\mathbf{w}_t^{\texttt{greedy}}$ | Prediction after $t$ steps under greedy decoding. |
| $\mathbf{w}_{t,N}^{\texttt{mv}}$ | Majority-vote prediction from $N$ sampled paths of length $t$. |
| $\Delta_t$ or $\Delta_T$ | Probability gap between the target coefficient and the most likely incorrect coefficient. |
| $p_{\texttt{trans}}$ | Lower bound on the transition probability toward the target state. |
| $p_{\texttt{recurr}}$ | Lower bound on the probability of remaining at the target state. |
| $\mu$ | Decay-rate parameter for the dependence on reasoning length. |
| $\alpha, \beta$ | Accuracy fitting parameters for sampling-number extrapolation. |
| $\gamma, \kappa$ | Fitting parameters for the probability-gap dependence on reasoning length. |

# C. Experiment Details

## C.1. Experiment Settings

**Details for Figure 1:**  We evaluate real-world LLM using the GSM8K dataset (Cobbe et al., 2021), employing SGLang (Zheng et al., 2024) as our inference framework; and use synthetic data with our theoretical framework to simulate practical decoding procedures. The experimental configurations are as follows:

- **LLM Performance on GSM8K with Varying Sample Number**: We employ Llama3.1-8b (Dubey et al., 2024) with an 8-shot chain-of-thought prompt following (Wei et al., 2022). For each question, we generate 256 potential answers using decoding temperature of 1.0. We implement an oracle reward model that perfectly validates answer correctness, and set the temperature to 0.0 for greedy search.

- **LLM Performance on GSM8K with Varying Reasoning Lengths**: Using Llama3.1-8b-instruct, we analyze performance across different reasoning lengths, defined as the token consumption per inference call. Following (Zhang & Chen, 2024), we incorporate token budgets into the prompts to constrain the model's responses. For each prompt, we generate 64 potential answers and create 10 random permutations of these answers. We define the reasoning length $T$ as the sum of token consumption across all prompts, and for multiple samples ($N > 1$), we average the token counts over $N$. The accuracy-tokens curves are plotted using transparent scattered points for individual permutations and fitted with trend lines. The prompt templates are provided in H.

- **IC-Linear Regression with Continuous Coefficients**: We configure the parameters as $n = 36, d = 72, \eta = 1 \times 10^{-3}, \sigma_\epsilon^2 = 1, \sigma^2 = 4$, and present results at gradient descent iterations $t = 950$.

- **IC-Linear Regression with Binary Coefficients**: We set the parameters to $n = 4, k = 1, d = 48, \eta = \frac{1}{4}, \sigma_\epsilon^2 = 0.25$.

**Details for Figure 2:**  we conduct numerical experiments on in-context linear regression with continuous coefficients (*above a-d*) and binary coefficients (*below e-h*), each setting we repeat 5 times, details are as follows:

- **Continuous case**: we set the parameters to $d = 72, n = 36, \eta = 10^{-3}$, and present results at gradient descent iterations $t = 950$.

- **Binary case**: In Figure 2 (e): we set $n = 40, k = 2, d = 30, \eta = \frac{1}{40}, \sigma_\epsilon = 0.1$; in (f): we set $n = 1, k = 1, d = 10, \eta = 1, \sigma_\epsilon = 0$; in (g): we set $n = 1, k = 1, d = 2, \eta = 1, \sigma_\epsilon = 0.1$; in (h): we set $n = 5, k = 1, d = 10, \eta = 1, \sigma_\epsilon = 0.1$.

- **Fitting accuracy with varying reasoning length** $T$: for $N = 1$, we fit the curve with

$$\texttt{Acc}(T, 1) \approx \alpha_1 - \beta_1 e^{-\mu_1 T},$$

for $N > 1$, we first approximate $\Delta_T \approx \texttt{Acc}(T, 1) \approx \alpha_1 - \beta_1 e^{-\nu_1 T}$, where $(\alpha_1, \beta_1, \nu_1)$ are obtained in case $N = 1$, then fit curve with

$$\texttt{Acc}(T, N) \approx \alpha_N - \beta_N e^{-\mu_N \Delta_T^2}.$$

**Details for Figure 3:**  We conduct experiments using GSM8K and a curated subset of the MATH dataset (Hendrycks et al., 2021), details are as follows:

- **MATH Dataset Subset**: We filter the MATH to extract problems at level 1 with integer answers, yielding a subset of 309 problems.

- We maintain consistent experimental settings with the GSM8K reasoning length evaluation as in Figure 1, utilizing Llama3.1-8b-instruct with a decoding temperature of 1.0. To facilitate the fitting process in Algorithm 4, we apply a scaling factor of $\frac{1}{10^5}$ to the token count, $T' = \frac{T}{10^5}$.

## C.2. Low-Cost-to-High Prediction algorithm

Our theoretical analysis reveals two critical terms $\mathcal{O}(e^{-\Delta_T^2 N/2})$ and $\mathcal{O}(e^{-\mu T})$ for the overall accuracy $\texttt{Acc}(T, N)$ and probability gap $\Delta_T$. These findings can provide valuable insights into real-world LLM inference.

To begin, we can observe that $\Delta_T$ changes with the number of reasoning steps $T$ in $\mathcal{O}(e^{-\mu T})$. This can be described as:

$$\Delta_T \approx \gamma - \kappa e^{-\mu T}. \tag{C.1}$$

Specifically, for sampling number of $N = 1$, here we *assume* we can directly express the overall accuracy as :

$$\texttt{Acc}(T, 1) \approx \gamma' - \kappa' e^{-\mu T}. \tag{C.2}$$

Note that Eq (C.2) and (C.1) shares the same $\mu$. To predict the final accuracy for a given sampling number $N$, here we introduce two additional parameters $(\alpha_{(T,N)}, \beta_{(T,N)})$ and formulate $\texttt{Acc}(T, N)$ as:

$$\texttt{Acc}(T, N) \approx \alpha_{(T,N)} - \beta_{(T,N)} e^{-\Delta_T^2 N/2}. \tag{C.3}$$

To effectively fit Eq (C.1) - (C.3), based on the results on Fig 2 (g) and (h), we further claim two conjectures:

- When $T$ is fixed, then Eq C.3 can be approximated by:

$$\texttt{Acc}(T, N) \approx \alpha_T - \beta_T e^{-\Delta_T^2 N/2}.$$

- When $N$ is fixed, then Eq C.3 can be approximated by:

$$\texttt{Acc}(T, N) \approx \alpha_N - \beta_N e^{-\Delta_T^2 N/2}.$$

This analysis enables us to predict model's high test-time computation performance using data from low-computation, resulting our Low-Cost-to-High Prediction Algorithm 4: , we validate our algorithm on GSM8K (Cobbe et al., 2021) and a subset of MATH (Hendrycks et al., 2021)

---

**Algorithm 4** Low-Cost-to-High Prediction algorithm

---

**Part 1:** Obtain $(\gamma, \kappa, \mu)$ in Eq C.1

1: **Input:** Data at varying cost $\{\texttt{Acc}^{(e)}(T_i, N_j)\}, T_i \in \mathcal{T}^{(e)}, N_j \in \mathcal{N}^{(e)}$;
2: $(\gamma', \kappa', \mu) \leftarrow$ Fit Eq C.2 with $\{\texttt{Acc}^{(e)}(T_i, 1)\}_{\mathcal{T}^{(e)}}$
3: $(\alpha_{T_1}, \beta_{T_1}, \Delta_{T_1}) \leftarrow$ Fit Eq C.2 with $\{\texttt{Acc}^{(e)}(T_1, N_j)\}$
4: $(\alpha_{T_2}, \beta_{T_2}, \Delta_{T_2}) \leftarrow$ Fit Eq C.2 with $\{\texttt{Acc}^{(e)}(T_2, N_j)\}$
5: $(\gamma, \kappa) \leftarrow$ Fit Eq C.1 with $\{(\Delta_{T_0}, \mu), (\Delta_{T_1}, \mu)\}$
6: **Return** $(\gamma, \kappa, \mu)$

**Part 2:** Predict accuracy with $(\gamma, \kappa, \mu)$ and low cost data

1: **Input:** $(\gamma, \kappa, \mu)$ in Eq C.1, $\mathcal{D}_N = \{\texttt{Acc}^{(e)}(T_1, N), \texttt{Acc}^{(e)}(T_2, N)\}$;
2: $\Delta_{T_i} \leftarrow \gamma - \kappa e^{-\mu T_i}, i = 1, 2$                                             {//Eq C.1}
3: $(\alpha_N, \beta_N) \leftarrow$ Fit Eq C.2 with two data points: $\{(\texttt{Acc}^{(e)}(T_1, N), P_{T_1}), (\texttt{Acc}^{(e)}(T_2, N), P_{T_2})\}$
4: Use Eq C.1, Eq C.2 with obtained $(\gamma, \kappa, \mu)$ and $(\alpha_N, \beta_N)$ to predict data with varying $T$.

---

The core ideal of Algorithm 4 is to first determine $(\gamma, \kappa, \mu)$ in Equation C.1. Subsequently, we can compute $\Delta_T$ and Equation C.2 using two additional parameters $\alpha_N, \beta_N$, obtainable from only two data points. Notably, since we use $\texttt{Acc}^{(e)}(T_0, N_j)$ and $\texttt{Acc}^{(e)}(T_1, N_j)$ during the initial parameter estimation (Algorithm 4 Part 1, lines 3-4), no additional data is required for subsequent predictions in part 2.

## D. Proofs for Section 3

### D.1. Proof of Proposition 4.3

*Proof of Proposition 4.3.* The proof is based on the proof of Theorem 3.2 of (Huang et al., 2025). We take the desired parameter $\theta_{\text{GD}} = \{\mathbf{V}_{\text{GD}}, \mathbf{W}_{\text{GD}}\}$ as following,

$$\mathbf{V}_{\text{GD}} := \begin{pmatrix} \mathbf{0} & \mathbf{0} & \mathbf{0} & \mathbf{0} \\ \mathbf{0} & \mathbf{0} & \mathbf{0} & \mathbf{0} \\ -\eta \cdot \mathbf{I}_d & \mathbf{0} & \mathbf{0} & \mathbf{0} \\ \mathbf{0} & \mathbf{0} & \mathbf{0} & \mathbf{0} \end{pmatrix}, \quad \mathbf{W}_{\text{GD}} := \begin{pmatrix} \mathbf{0} & \mathbf{0} & \mathbf{I}_d & \mathbf{0} \\ \mathbf{0} & \mathbf{0} & \mathbf{0} & -1 \\ \mathbf{0} & \mathbf{0} & \mathbf{0} & \mathbf{0} \\ \mathbf{0} & \mathbf{0} & \mathbf{0} & \mathbf{0} \end{pmatrix},$$

Then one can check that when inputting $\mathbf{H}_\ell$ in the form of

$$\mathbf{H}_\ell = \begin{pmatrix} \mathbf{x}_1 & \cdots & \mathbf{x}_n & \mathbf{0} & \cdots & \mathbf{0} \\ y_1 & \cdots & y_n & 0 & \cdots & 0 \\ \mathbf{0} & \cdots & \mathbf{0} & \mathbf{w}_0 & \cdots & \mathbf{w}_\ell \\ 0 & \cdots & 0 & 1 & \cdots & 1 \end{pmatrix},$$

the output embedding of the transformer at the last token is given by

$$(\widetilde{\mathbf{H}}_\ell)_{:,-1} = \begin{pmatrix} \mathbf{0} \\ 0 \\ \widehat{\mathbf{w}}_\ell \\ 1 \end{pmatrix}, \quad \widetilde{\mathbf{w}}_\ell = \mathbf{w}_\ell - \frac{\eta}{n} \cdot \mathbf{X}^\top (\mathbf{X}\mathbf{w}_\ell - \mathbf{y}).$$

Thus if we take the sampling algorithm $\texttt{Sampling\_Alg}(\cdot)$ satisfying the form of

$$\texttt{Sampling\_Alg}(\mathbf{h}) = \delta_{\mathbf{0}}(\cdot) \otimes \delta_0(\cdot) \otimes p\left(\cdot|(\mathbf{h})_{d+2:2d+1}\right) \otimes \delta_1(\cdot),$$

for some conditional distribution $p : \mathbb{R}^d \mapsto \mathcal{P}(\mathbb{R}^d)$, then the embedding of the next token would be

$$\mathbf{h}_{\ell+1} = \begin{pmatrix} \mathbf{0} \\ 0 \\ \mathbf{w}_\ell \\ 1 \end{pmatrix}, \quad \mathbf{w}_{\ell+1} \sim p\left(\cdot \left| \mathbf{w}_\ell - \frac{\eta}{n} \cdot \mathbf{X}^\top (\mathbf{X}\mathbf{w}_\ell - \mathbf{y})\right.\right),$$

by Algorithm 1. Iterating the above argument from $\ell = 0$ to $t - 1$ completes the proof of Proposition 4.3. $\qquad\square$

### D.2. Special Case: Vanilla Multi-step Grandient Descent with CoT

One special case of Proposition 4.3 is a transformer that explicitly performs standard multi-step gradient descent (GD) (Huang et al., 2025), i.e., $p(\cdot|x) = \delta_x(\cdot)$, so that the final prediction of the regression coefficient after $t$ reasoning steps is given by

$$\mathbf{w}_{\text{GD}} := (\mathbf{H}_t)_{d+2:2d+1,n+t} = \left(\mathbf{I}_d - \left(\mathbf{I}_d - \frac{\eta}{n} \cdot \mathbf{X}^\top \mathbf{X}\right)^t\right) \mathbf{X}^\top (\mathbf{X}\mathbf{X}^\top)^{-1} \mathbf{y}. \tag{D.1}$$

We note that (Huang et al., 2025) considers transformer CoT reasoning for in-context-linear regression with *noiseless* labels, but here we allow the existence of label noise.

## E. Theoretical Analysis in Section 5 Continued

**Theorem E.1** (Excess risk of vanilla multi-step GD with CoT: general covariance matrix). *Under Assumption F.1, taking the step size $\eta \le \|\mathbf{H}\|_2^{-1}$ and CoT length $t$, with probability at least $1 - 1/\text{poly}(n)$, it holds that*

$$\mathbb{E}_{\epsilon, \mathbf{w}^*}\left[\mathcal{E}(\mathbf{w}_{\text{GD}})\right] \lesssim \omega^2 \cdot \left(\frac{\widetilde{\lambda}^2}{n^2} \cdot \sum_{1 \le i \le k^*} \frac{1}{\lambda_i} + \sum_{k^* < i \le d} \lambda_i\right) + \sigma_\epsilon^2 \cdot \left(\frac{k^*}{n} + \frac{n}{\widetilde{\lambda}^2} \cdot \sum_{k^* < i \le d} \lambda_i^2\right),$$

*where the quantities are as follows*

$$k^* := \min\left\{k : n\lambda_{k+1} \le \frac{n}{\eta t} + \sum_{k < i \le d} \lambda_i\right\}, \quad \widetilde{\lambda} := \frac{n}{\eta t} + \sum_{k^* < i \le d} \lambda_i. \tag{E.1}$$

*Proof of Theorem E.1.* Please refer to Appendix F.1 for a proof of Theorem E.1. $\qquad\square$

**Theorem E.2** (Excess risk of noisy multi-step noisy GD with CoT and ensembling). *Under Assumption F.1, taking the step size $\eta \le \|\mathbf{H}\|_2^{-1}$ and CoT length $t$, we have the following risk bounds for $\mathbf{w}_{\text{avg}}$.*

1. *Constant noise transformation function (Example 4.1): with probability at least $1 - 1/\text{poly}(n)$,*

$$\mathbb{E}_{\epsilon, \mathbf{w}^*}\left[\mathcal{E}(\mathbf{w}_{\text{GD}})\right] \lesssim \omega^2 \cdot \left(\frac{\widetilde{\lambda}^2}{n^2} \cdot \sum_{1 \le i \le k^*} \frac{1}{\lambda_i} + \sum_{k^* < i \le d} \lambda_i\right) + \frac{\vartheta_{n,t}}{N},$$

*where the quantities $k^*$ and $\widetilde{\lambda}$ are defined as the same as (E.1), and $\vartheta_t$ is defined as*

$$\vartheta_{n,t} := \sigma^2 d \cdot \left(t \cdot \sqrt{\frac{r(\mathbf{H}) \vee \log(\text{poly(n)})}{n}} + \frac{1}{\eta}\right), \tag{E.2}$$

*with $r(\mathbf{H}) = \text{Tr}(\mathbf{H})/\|\mathbf{H}\|_2$ being the effective rank of $\mathbf{H}$.*

2. *Linear noise transformation function (Example 4.2): taking the noise variance $\sigma^2 \asymp d^{-1}$ and the reasoning path length $t > \sigma^{-2} \cdot \log 2$, with probability at least $1 - 1/\text{poly}(n)$,*

$$\mathbb{E}_{\epsilon, \mathbf{w}^*, \boldsymbol{\xi}}\left[\mathcal{E}(\mathbf{w}_{\text{avg}})\right] \lesssim \omega^2 \cdot \left(\frac{(\widetilde{\lambda}^{\text{Bias}})^2}{n^2} \cdot \sum_{1 \le i \le k^*_{\text{Bias}}} \frac{1}{\lambda_i} + \sum_{k^*_{\text{Bias}} < i \le d} \lambda_i\right) + \sigma_\epsilon^2 \cdot \left(\frac{k^*_{\text{Var}}}{n} + \frac{n}{(\widetilde{\lambda}^{\text{Var}})^2} \cdot \sum_{k^*_{\text{Var}} < i \le d} \lambda_i^2\right) + \frac{\varsigma_n}{N},$$

*where the quantities $\widetilde{\lambda}^{\text{Bias}}$, $\widetilde{\lambda}^{\text{Var}}$, $k^*_{\text{Bias}}$, and $k^*_{\text{Var}}$ are defined as following respectively,*

$$k^*_{(\diamond)} := \min\left\{k \in [d] : n\lambda_{k+1} \le \widetilde{\lambda}^{(\diamond)}_{\text{effect}} + \sum_{k < i \le d} \lambda_i\right\}, \quad \widetilde{\lambda}^{(\diamond)} := \widetilde{\lambda}^{(\diamond)}_{\text{effect}} + \sum_{k^* < i \le d} \lambda_i, \quad \text{for } (\diamond) \in \{\text{Bias}, \text{Var}\},$$

*with $\widetilde{\lambda}_{\text{effect}}^{\text{Bias}}$ and $\widetilde{\lambda}_{\text{effect}}^{\text{Var}}$ defined as,*

$$\widetilde{\lambda}_{\text{effect}}^{\text{Bias}} := \frac{n}{\eta} \cdot \left( \frac{2}{t} + \frac{\sigma^2}{1 - \sigma^2} \left( 1 + \frac{2}{t} \right) \right), \quad \widetilde{\lambda}_{\text{effect}}^{\text{Var}} := \frac{\sigma^2 n}{(1 - \sigma^2)\eta},$$

*and the quantity $\varsigma_n$, is given by*

$$\varsigma_n := \left( \frac{\eta \sigma_\epsilon^2 d}{n \sigma^2} \cdot \text{Tr}(\mathbf{H}) + \omega^2 \right) \cdot \|\mathbf{H}\|_2. \tag{E.3}$$

*Proof of Theorem E.2.* Please refer to Appendix F.3 for a proof of Theorem E.2. $\square$

**Corollary E.3** (Theorem 5.2 restated)**.** *Under the same assumptions and setups as in Theorem E.2, additionally assuming that the spectrum of $\mathbf{H}$ satisfies polynomially decaying, i.e., $\lambda_i = i^{-(r+1)}$ for some constant $r \geq 0$, we have the following results.*

1. *Constant noise transformation function (Example 4.1): taking the reasoning path length $t \lesssim \eta(r+1)^{(r+2)/2} n^{(r+1)/2}$ and the sampling path number*

$$N \geq N_c := \left( \sigma^2 d \cdot \left( t \cdot \sqrt{\frac{r(\mathbf{H}) \vee \log(\text{poly}(n))}{n}} + \frac{1}{\eta} \right) \right) \cdot \left( \omega^2 \cdot \left( \frac{1}{t\eta} \right)^{\frac{r}{r+1}} + \frac{\sigma_\epsilon^2}{n} \cdot (t\eta)^{\frac{1}{r+1}} \right)^{-1}, \tag{E.4}$$

*then with probability at least $1 - 1/\text{poly}(n)$,*

$$\mathbb{E}\left[ \mathcal{E}(\mathbf{w}_{\text{avg}}) \right] \lesssim \omega^2 \cdot \left( \frac{1}{t\eta} \right)^{\frac{r}{r+1}} + \frac{\sigma_\epsilon^2}{n} \cdot (t\eta)^{\frac{1}{r+1}}.$$

2. *Linear noise transformation function (Example 4.2): taking the noise variance $\sigma^2 \asymp d^{-1}$, the reasoning path length $\sigma^{-2} \cdot \log 2 < t$, and the sampling path number*

$$N \geq N_l := \left( \omega^2 + \frac{\eta \sigma_\epsilon^2 d \cdot \text{Tr}(\mathbf{H})}{n \sigma^2} \right) \cdot \|\mathbf{H}\|_2 \cdot \left( \omega^2 \cdot \left( \frac{\sigma^2}{\eta \cdot (1 - \sigma^2)} \right)^{\frac{r}{r+1}} + \frac{\sigma_\epsilon^2}{n} \cdot \left( \frac{\eta \cdot (1 - \sigma^2)}{\sigma^2} \right)^{\frac{1}{r+1}} \right)^{-1}$$

$$\asymp \left( \omega^2 + \frac{\sigma_\epsilon^2}{n} \cdot \eta d^2 \right) \cdot \left( \omega^2 \cdot \left( \frac{1}{\eta d} \right)^{\frac{r}{r+1}} + \frac{\sigma_\epsilon^2}{n} \cdot (\eta d)^{\frac{1}{r+1}} \right)^{-1} \tag{E.5}$$

*then with probability at least $1 - 1/\text{poly}(n)$,*

$$\mathbb{E}\left[ \mathcal{E}(\mathbf{w}_{\text{avg}}) \right] \lesssim \omega^2 \cdot \widetilde{\lambda}^{\frac{r}{r+1}} + \frac{\sigma_\epsilon^2}{n} \cdot \left( \frac{\eta(1 - \sigma^2)}{\sigma^2} \right)^{\frac{1}{r+1}},$$

*where $\widetilde{\lambda} := \eta^{-1}(2t^{-1} + \sigma^2(1 + 2t^{-1})/(1 - \sigma^2))$.*

*Here the expectation is taken with respect to $\epsilon$, $\mathbf{w}^*$, and the sampling noise $\boldsymbol{\xi}$ across different reasoning steps and paths.*

**Remark E.4.** *Under the parameter regime of (5.1), i.e.,*

$$\omega \asymp 1, \quad \sigma_\epsilon \asymp 1, \quad n \asymp \eta d, \quad \sigma^2 \asymp d^{-1},$$

*we can obtain further simplifications of the above result. Concretely, for the linear NFT setup, the number of sample paths needed is given by*

$$N \geq N_l \asymp (\omega^2 + \sigma_\epsilon^2 d) \cdot \left( (\omega^2 + \sigma_\epsilon^2) \cdot \left( \frac{1}{\eta d} \right)^{\frac{r}{r+1}} \right)^{-1} \asymp d^{\frac{2r+1}{r+1}},$$

*and the excess risk bound is explicitly calculated by*

$$\mathbb{E}_{\boldsymbol{\epsilon}, \mathbf{w}^*, \boldsymbol{\xi}} \left[ \mathcal{E}(\mathbf{w}_{\texttt{avg,linear}}) \right] \lesssim \left( \omega^2 + \sigma_\epsilon^2 \right) \cdot (\eta d)^{-\frac{r}{r+1}} \asymp d^{-\frac{r}{r+1}}.$$

*In contrast, we can also calculate that the risk bounds for either GD or ensemble with constant NFT is then given by*

$$\mathbb{E}_{\boldsymbol{\epsilon}, \mathbf{w}^*} \left[ \mathcal{E}(\mathbf{w}_{\texttt{GD}}) \right], \mathbb{E}_{\boldsymbol{\epsilon}, \mathbf{w}^*, \boldsymbol{\xi}} \left[ \mathcal{E}(\mathbf{w}_{\texttt{avg,const}}) \right] \lesssim \widetilde{t}^{\frac{1}{r+1}} \cdot \left( \omega^2 + \sigma_\epsilon^2 \right) \cdot (\eta d)^{-\frac{r}{r+1}} \asymp \widetilde{t}^{\frac{1}{r+1}} \cdot d^{-\frac{r}{r+1}}.$$

*where $\widetilde{t} = \sigma^2 \cdot t$ is the scaled reasoning length, satisfying $\widetilde{t} \lesssim d^{(r-1)/2}$.*

# F. Proofs for In-context Linear Regression with Continuous Coefficient (Section 5)

We denote the sample covariance matrix of the in-context data as $\boldsymbol{\Sigma} := n^{-1} \mathbf{X}^\top \mathbf{X} \in \mathbb{R}^{d \times d}$, and we define the gram matrix of the in-context data as $\mathbf{A} := \mathbf{X}\mathbf{X}^\top \in \mathbb{R}^{n \times n}$. Our results in this section depend on the following standard technical assumptions on the in-context data and task distributions.

**Assumption F.1** (Data distribution). *We assume the following on the in-context data distribution $\mathcal{D}_{\mathbf{w}^*}$:*

1. *The columns of $\mathbf{H}^{-1/2}\mathbf{x}$ are independent and $1$-subGaussian;*

2. *The labels are generated according to $y = \mathbf{x}^\top \mathbf{w}^* + \epsilon$, where the label noise $\epsilon$ is independent of $\mathbf{x}$ and satisfies $\mathbb{E}[\epsilon] = 0$ and $\mathbb{E}[\epsilon^2] = \sigma_\epsilon^2$ for some constant $\sigma_\epsilon > 0$;*

3. *The true coefficient $\mathbf{w}^*$ follows the Gaussian prior, i.e., $\mathbf{w}^* \sim \mathcal{N}(\mathbf{0}, \omega^2 \cdot \mathbf{I}_d)$ for some constant $\omega > 0$.*

## F.1. Proof of Theorem E.1

*Proof of Theorem E.1.* This follows from the same arguments as in the proof of Theorem 4.3 in (Zou et al., 2022). We refer the readers to their proofs for seek of simplicity. $\qquad\square$

## F.2. Proof of Proposition 5.1

*Proof of Proposition 5.1.* As a special case of Theorem E.1, we begin by figuring out the optimal index $k^*$. We are going to prove that under the conditions in Proposition 5.1, the optimal index is given by

$$k^* = (\eta t)^{\frac{1}{r+1}} - 1.$$

Notice that here without loss of generality we assume that the above quantity is an integer since otherwise we can twist $\eta$ (which is continuous) a little bit to make it an integer. And also we notice that the above $k^* \leq d$ due to our condition on $t$ in Proposition 5.1. To prove this, it suffices to check that the above $k^*$ is the smallest one satisfying the constraint in (E.1). To show it satisfies the constraint, consider

$$n\lambda_{k^*+1} = \frac{n}{(k^*+1)^{r+1}} = \frac{n}{\eta t} \leq \frac{n}{\eta t} + \sum_{k < i \leq d} \lambda_i.$$

To show that it is the smallest one satisfying the constraint, let's consider the other side of the inequality for $k^* - 1$. We have the following calculations. On the one hand, we have

$$n\lambda_{k^*} = \frac{n}{\left( (\eta t)^{\frac{1}{r+1}} - 1 \right)^{r+1}} = \frac{n}{\eta t} \cdot \frac{1}{\left( 1 - (\eta t)^{-\frac{1}{r+1}} \right)^{r+1}} \geq \frac{n}{\eta t} \cdot \left( 1 + (r+1) \cdot \left( \frac{1}{\eta t} \right)^{\frac{1}{r+1}} \right), \tag{F.1}$$

where the last inequality follows using $\log(1 + x) \leq x$ and $\exp(x) \geq 1 + x$ to obtain the following argument

$$\frac{1}{\left( 1 - (\eta t)^{-\frac{1}{r+1}} \right)^{r+1}} = \exp\left( -(r+1) \log\left( 1 - (\eta t)^{-\frac{1}{r+1}} \right) \right) \geq \exp\left( (r+1)(\eta t)^{-\frac{1}{r+1}} \right) \geq 1 + (r+1)(\eta t)^{-\frac{1}{r+1}}.$$

On the other hand, we have that

$$\frac{n}{\eta t} + \sum_{k^*-1 < i \leq d} \lambda_i \leq \frac{n}{\eta t} + \sum_{i > k^*-1} \frac{1}{i^{r+1}} \leq \frac{n}{\eta t} + \frac{1}{\left((\eta t)^{\frac{1}{r+1}} - 1\right)^r} \lesssim \frac{n}{\eta t} + \left(\frac{1}{\eta t}\right)^{\frac{r}{r+1}}. \tag{F.2}$$

Now to see that $k^* - 1$ does not satisfies the constraint, in view of (F.1) and (F.2), it boils down to show that

$$\frac{n}{\eta t} \cdot \left(1 + (r+1) \cdot \left(\frac{1}{\eta t}\right)^{\frac{1}{r+1}}\right) \geq \frac{n}{\eta t} + \left(\frac{1}{\eta t}\right)^{\frac{r}{r+1}},$$

which is equivalent to restricting the reasoning path length $t$ satisfying $t \leq \eta \cdot (r+1)^{\frac{r+1}{2}} \cdot n^{\frac{r+1}{2}}$. According to our condition on the reasoning path length $t$ in Proposition 5.1, this requirement does hold, and thus $k^* - 1$ does not satisfy the constraint. Therefore we have proved that $k^* = (\eta t)^{\frac{1}{r+1}} - 1$.

With the $k^*$ in hand, we can then follow the same arguments as in the proof of Corollary 4.5 in (Zou et al., 2022) to obtain the final result. This completes the proof of Proposition 5.1. $\qquad\square$

## F.3. Proof of Theorem E.2

### F.3.1. PROOF FOR EXAMPLE 4.1

*Proof of Theorem E.2 for Example 4.1.* Under this setting, each reasoning path is generated though the following iteration:

$$\mathbf{w}_{t+1}^{(j)} = \mathbf{w}_t^{(j)} - \frac{\eta}{n} \mathbf{X}^\top (\mathbf{X}\mathbf{w}_t^{(j)} - \mathbf{y}) + \boldsymbol{\xi}_t^{(j)}.$$

Based on this, we define the expected path $\mathbf{w}_t^{\mathrm{GD}(\eta;\mathbf{X},\mathbf{y})}$ and the fluctuation $\Delta_t^{(j)}$ iteratively as

$$\mathbf{w}_{t+1}^{\mathrm{GD}(\eta;\mathbf{X},\mathbf{y})} = \mathbf{w}_t^{\mathrm{GD}(\eta;\mathbf{X},\mathbf{y})} - \frac{\eta}{n} \mathbf{X}^\top (\mathbf{X}\mathbf{w}_t^{\mathrm{GD}(\eta;\mathbf{X},\mathbf{y})} - \mathbf{y}),$$

$$\Delta_{t+1}^{(j)} = \mathbf{w}_t^{(j)} - \mathbf{w}_t^{\mathrm{GD}(\eta;\mathbf{X},\mathbf{y})}$$

$$= (\mathbf{I} - \eta\boldsymbol{\Sigma})\Delta_t^{(j)} + \boldsymbol{\xi}_t^{(j)}.$$

By this characterization, we see that $\{\Delta_t^{(j)}\}_{j \leq N}$ is a sequence of iid zero-mean random variable for fixed $t$. This expectation-fluctuation decomposition allows us to recast the risk of the sample averaged output as

$$\mathcal{E}(\mathbf{w}_t^{\mathrm{avg}}) = \mathcal{E}(\mathbf{w}_t^{\mathrm{GD}(\eta;\mathbf{X},\mathbf{y})}) + N^{-1}\mathbb{E}\big[\|\Delta_t^{(1)}\|_{\mathbf{H}}^2\big]. \tag{F.3}$$

In Theorem E.1, we have characterized the average-case risk of the gradient descent, therefore it suffices to study the fluctuation of a single reasoning path. In the sequel, we drop the superscript $j$ for simplicity. Define $\mathbf{S}_t = \mathbb{E}[\Delta_t \Delta_t^\top]$, then we have that

$$\mathbf{S}_{t+1} = (\mathbf{I} - \eta\boldsymbol{\Sigma})\mathbf{S}_t(\mathbf{I} - \eta\boldsymbol{\Sigma})^\top + \sigma^2 \mathbf{I}$$

$$= \sum_{j=0}^t \sigma^2 (\mathbf{I} - \eta\boldsymbol{\Sigma})^{2j},$$

where the last identity holds because of the deterministic initialization $\mathbf{S}_0 = 0$. Now we have that

$$\mathbb{E}[\|\Delta_t^{(j)}\|_{\mathbf{H}}^2] = \langle \mathbf{S}_t, \boldsymbol{\Sigma} \rangle + |\langle \mathbf{S}_t, \mathbf{H} - \boldsymbol{\Sigma} \rangle|$$

$$\leq \mathrm{Tr}\Big(\sum_{j=0}^{t-1} \sigma^2 (\mathbf{I} - \eta\boldsymbol{\Sigma})^{2j}\boldsymbol{\Sigma}\Big) + \mathrm{Tr}(\mathbf{S}_t) \cdot \|\mathbf{H} - \boldsymbol{\Sigma}\|_2. \tag{F.4}$$

For the first term above, we have that $\sum_{j=0}^t (1 - \eta\lambda)^{2j}\lambda \leq 1/\eta$ for $\lambda \in [0, 1/\eta]$. For the second term , we have by Koltchinskii & Lounici (2017, Theorem 9) that there exists an event with probability $1 - \delta$ over the randomness of $\mathbf{X}$, on which it holds that

$$\|\mathbf{H} - \boldsymbol{\Sigma}\|_2 \lesssim \sqrt{\frac{r(\mathbf{H}) \vee \log(1/\delta)}{n}},$$

where $r(\mathbf{H}) = \mathrm{Tr}(\mathbf{H})/\|\mathbf{H}\|_2$ is the effective rank of $\mathbf{H}$. And we have the trivial upper bound that $\mathrm{Tr}(\mathbf{S}_t) \le \sigma^2 d \cdot t$. Plugging them into (F.3) and (F.4), we get that

$$\mathcal{E}(\mathbf{w}_t^{\mathrm{avg}}) \le \mathcal{E}(\mathbf{w}_t^{\mathrm{GD}(\eta;\mathbf{X},\mathbf{y})}) + N^{-1}\langle \mathbf{S}_t, \mathbf{H}\rangle$$

$$\le \mathcal{E}(\mathbf{w}_t^{\mathrm{GD}(\eta;\mathbf{X},\mathbf{y})}) + \frac{\sigma^2 d}{N}\left(t \cdot \sqrt{\frac{r(\mathbf{H}) \vee \log(1/\delta)}{n}} + \frac{1}{\eta}\right).$$

This concludes the proof of the theorem. □

### F.3.2. PROOF FOR EXAMPLE 4.2

Now we give the proof of Theorem E.2 for Example 4.2. The proof relies on the following key lemmas.

**Lemma F.2** (Error decomposition). *The difference between $\mathbf{w}_{\mathrm{avg}}$ and the true coefficient $\mathbf{w}^*$ can be decomposed as following,*

$$\left\|\mathbf{w}_{\mathrm{avg}} - \mathbf{w}^*\right\|_{\mathbf{H}}^2 \le \mathrm{Bias} + \mathrm{Variance} + \mathrm{Fluctuation},$$

*where each of the three terms are defined as following,*

$$\mathrm{Bias} := \left\|\left(\mathbf{X}^\top \mathbf{G}^{-1}\mathbf{X} - \mathbf{I}_d\right)\mathbf{w}^*\right\|_{\mathbf{H}}^2, \quad \mathrm{Variance} = \left\|\mathbf{X}^\top \mathbf{G}^{-1}\boldsymbol{\epsilon}\right\|_{\mathbf{H}}^2, \quad \mathrm{Fluctuation} = \left\|\frac{1}{N}\sum_{j=1}^N \Delta^{(j)}\right\|_{\mathbf{H}}^2, \quad (\mathrm{F.5})$$

*with the matrix $\mathbf{G} \in \mathbb{R}^{n \times n}$ and the vectors $\{\Delta^{(j)}\}_{j=1}^N$ defined as following,*

$$\mathbf{G} := \left(\frac{\sigma^2 n}{(1-\sigma^2)\eta}\cdot \mathbf{I}_n + \mathbf{A}\right)\left(\mathbf{I}_n - \left(1-\sigma^2\right)^t \cdot \left(\mathbf{I}_n - \frac{\eta}{n}\cdot \mathbf{A}\right)^t\right)^{-1}, \quad (\mathrm{F.6})$$

$$\Delta^{(j)} := \sum_{k=0}^{t-1}\left(\prod_{\ell=0}^{k-1}\left(\mathbf{I}_d - \boldsymbol{\xi}_{t-\ell}^{(j)}(\boldsymbol{\xi}_{t-\ell}^{(j)})^\top\right)\left(\mathbf{I}_d - \eta\boldsymbol{\Sigma}\right)\right)\left(\mathbf{I}_d - \boldsymbol{\xi}_{t-k}^{(j)}(\boldsymbol{\xi}_{t-k}^{(j)})^\top\right)\cdot \frac{\eta}{n}\cdot \mathbf{X}^\top\mathbf{y}$$

$$- \sum_{k=0}^{t-1}\left(1-\sigma^2\right)^{k+1}\left(\mathbf{I}_d - \eta\boldsymbol{\Sigma}\right)^k \cdot \frac{\eta}{n}\cdot \mathbf{X}^\top\mathbf{y}.$$

*Proof of Lemma F.2.* By definition, the output $\mathbf{w}_{\mathrm{avg}}$ is defined as

$$\mathbf{w}_{\mathrm{avg}} := \frac{1}{N}\sum_{j=1}^N \mathbf{w}_t^{(j)}, \quad (\mathrm{F.7})$$

where for each $j \in [N]$, the coefficient $\mathbf{w}_t^{(j)}$ is given by

$$\mathbf{w}_t^{(j)} = \sum_{k=0}^{t-1}\left(\prod_{\ell=0}^{k-1}\left(\mathbf{I}_d - \boldsymbol{\xi}_{t-\ell}^{(j)}(\boldsymbol{\xi}_{t-\ell}^{(j)})^\top\right)\left(\mathbf{I}_d - \eta\boldsymbol{\Sigma}\right)\right)\left(\mathbf{I}_d - \boldsymbol{\xi}_{t-k}^{(j)}(\boldsymbol{\xi}_{t-k}^{(j)})^\top\right)\cdot \frac{\eta}{n}\cdot \mathbf{X}^\top\mathbf{y}$$

$$= \Delta^{(j)} + \underbrace{\sum_{k=0}^{t-1}\left(1-\sigma^2\right)^{k+1}\left(\mathbf{I}_d - \eta\boldsymbol{\Sigma}\right)^k \cdot \frac{\eta}{n}\cdot \mathbf{X}^\top\mathbf{y}}_{:=\mathbf{w}_t}.$$

Now we decompose the difference between $\mathbf{w}_{\mathrm{avg}}$ in (F.7) and the truth $\mathbf{w}^*$ as following, considering

$$\mathbf{w}_{\mathrm{avg}} - \mathbf{w}^* = \frac{1}{N}\sum_{j=1}^N \mathbf{w}_t^{(j)} - \mathbf{w}^* = \mathbf{w}_t - \mathbf{w}^* + \frac{1}{N}\sum_{j=1}^N \Delta^{(j)}, \quad (\mathrm{F.8})$$

where the difference $\mathbf{w}_t - \mathbf{w}^*$ can be further explicitly expanded as

$$
\begin{aligned}
\mathbf{w}_t - \mathbf{w}^* &= \sum_{k=0}^{t-1} (1-\sigma^2)^{k+1} (\mathbf{I}_d - \eta\boldsymbol{\Sigma})^k \cdot \frac{\eta}{n} \cdot \mathbf{X}^\top \mathbf{y} - \mathbf{w}^* \\
&= \sum_{k=0}^{t-1} (1-\sigma^2)^{k+1} (\mathbf{I}_d - \eta\boldsymbol{\Sigma})^k \cdot \frac{\eta}{n} \cdot \mathbf{X}^\top (\mathbf{W}\mathbf{w}^* + \boldsymbol{\epsilon}) - \mathbf{w}^* \\
&= (1-\sigma^2) \cdot \left(\mathbf{I}_d - (1-\sigma^2)^t (\mathbf{I}_d - \eta\boldsymbol{\Sigma})^t\right) \left(\sigma^2 \mathbf{I}_d + (1-\sigma^2)\eta\boldsymbol{\Sigma}\right)^{-1} \cdot \frac{\eta}{n} \cdot \mathbf{X}^\top \mathbf{X}\mathbf{w}^* - \mathbf{w}^* \\
&\quad + (1-\sigma^2) \cdot \left(\mathbf{I}_d - (1-\sigma^2)^t (\mathbf{I}_d - \eta\boldsymbol{\Sigma})^t\right) \left(\sigma^2 \mathbf{I}_d + (1-\sigma^2)\eta\boldsymbol{\Sigma}\right)^{-1} \cdot \frac{\eta}{n} \cdot \mathbf{X}^\top \mathbf{X}\boldsymbol{\epsilon} \\
&= \left(\mathbf{X}^\top \mathbf{G}^{-1} \mathbf{X} - \mathbf{I}_d\right)\mathbf{w}^* + \mathbf{X}^\top \mathbf{G}^{-1}\boldsymbol{\epsilon},
\end{aligned}
\tag{F.9}
$$

where the last equality uses the definition of the matrix $\mathbf{G}$ in (F.6) and the fact that

$$
\begin{aligned}
&\left(\mathbf{I}_d - (1-\sigma^2)^t (\mathbf{I}_d - \eta\boldsymbol{\Sigma})^t\right)\left(\sigma^2 \mathbf{I}_d + (1-\sigma^2)\eta\boldsymbol{\Sigma}\right)^{-1} \mathbf{X}^\top \\
&= \mathbf{X}^\top \left(\mathbf{I}_n - (1-\sigma^2)^t \left(\mathbf{I}_d - \frac{\eta}{n}\mathbf{A}\right)^t\right)\left(\sigma^2 \mathbf{I}_n + (1-\sigma^2)\eta\mathbf{A}\right)^{-1}.
\end{aligned}
$$

Finally, by combining (F.8) and (F.9), we can arrive at

$$
\left\|\mathbf{w}_{\text{avg}} - \mathbf{w}^*\right\|_{\mathbf{H}}^2 = \left\|\left(\mathbf{X}^\top \mathbf{G}^{-1} \mathbf{X} - \mathbf{I}_d\right)\mathbf{w}^* + \mathbf{X}^\top \mathbf{G}^{-1}\boldsymbol{\epsilon} + \frac{1}{N}\sum_{j=1}^{N} \Delta^{(j)}\right\|_{\mathbf{H}}^2 \leq \text{Bias} + \text{Variance} + \text{Fluctuation}.
$$

This completes the proof of Lemma F.2. $\qquad\square$

**Lemma F.3.** *The matrix $\mathbf{G}$ satisfies the that for any CoT length $t \geq \sigma^{-2} \cdot \log 2$, it holds that*

$$
\frac{\sigma^2 n}{(1-\sigma^2)\eta} \cdot \mathbf{I}_n + \mathbf{A} \preceq \mathbf{G} \preceq \frac{n}{\eta} \cdot \left(\frac{2}{t} + \frac{\sigma^2}{1-\sigma^2}\left(1+\frac{2}{t}\right)\right) \cdot \mathbf{I}_n + \mathbf{A}.
$$

*Proof of Lemma F.3.* It is direct from the definition of $\mathbf{G}$ in (F.6) to see the left side of the inequality. To prove the right side of the inequality, consider that by (F.6), we have the following,

$$
\mathbf{G} - \left(\frac{\sigma^2 n}{(1-\sigma^2)\eta} \cdot \mathbf{I}_n + \mathbf{A}\right) \tag{F.10}
$$
$$
= (1-\sigma^2)^t \cdot \left(\frac{\sigma^2 n}{(1-\sigma^2)\eta} \cdot \mathbf{I}_n + \mathbf{A}\right)\left(\mathbf{I}_n - \frac{\eta}{n} \cdot \mathbf{A}\right)^t \left(\mathbf{I}_n - (1-\sigma^2)^t \cdot \left(\mathbf{I}_n - \frac{\eta}{n} \cdot \mathbf{A}\right)^t\right)^{-1}.
$$

To proceed, it suffices to consider the real-valued single-variable function $f$ defined as

$$
f(x) = \frac{\left(\eta^{-1}(1-\sigma^2)^{-1} n\sigma^2 + x\right) \cdot \left(1 - n^{-1}\eta x\right)^t}{1 - (1-\sigma^2)^t \cdot \left(1 - n^{-1}\eta x\right)^t}.
$$

On the one hand, for $t \geq \sigma^{-2} \cdot \log 2$, we have $t > -\log 2 / \log(1-\sigma^2)(1-n^{-1}\eta x)$, and thus

$$
1 - (1-\sigma^2)^t \cdot \left(1 - n^{-1}\eta x\right)^t \geq \frac{1}{2}. \tag{F.11}
$$

On the other hand, by direct calculations we can see that the numerator is upper bounded by

$$
\left(\frac{\sigma^2 n}{(1-\sigma^2)\eta} + x\right) \cdot \left(1 - n^{-1}\eta x\right)^t \leq \frac{1}{t} \cdot \frac{n}{\eta} \cdot \left(\frac{\sigma^2}{1-\sigma^2} + 1\right). \tag{F.12}
$$

Consequently, by combining (F.11) and (F.12), we can see that for $t \geq \sigma^{-2} \cdot \log 2$,

$$f(x) \leq \frac{2}{t} \cdot \frac{n}{\eta} \cdot \left( \frac{\sigma^2}{1 - \sigma^2} + 1 \right),$$

which, combined with (F.10), further indicates that

$$\mathbf{G} - \left( \frac{\sigma^2 n}{(1 - \sigma^2)\eta} \cdot \mathbf{I}_n + \mathbf{A} \right) \preceq \frac{2}{t} \cdot \frac{n}{\eta} \cdot \left( \frac{\sigma^2}{1 - \sigma^2} + 1 \right) \cdot \mathbf{A}.$$

This completes the proof of the right side inequality of Lemma F.3 and finishes the proof. $\square$

**Lemma F.4** (Bias error). *Under Assumption F.1, taking the step size $\eta \lesssim \mathrm{Tr}(\mathbf{H})^{-1}$ and for any $k \in [d]$, with probability at least $1 - 1/\mathrm{poly}(n)$, it holds that*

$$\mathbb{E}_{\mathbf{w}^*}[\mathrm{Bias}] \lesssim \omega^2 \cdot \left( \frac{1}{n^2} \cdot \left( \frac{n}{\eta} \cdot \left( \frac{2}{t} + \frac{\sigma^2}{1 - \sigma^2} \cdot \left( 1 + \frac{2}{t} \right) \right) + \sum_{k < i \leq d} \lambda_i \right)^2 \cdot \sum_{1 \leq i \leq k} \frac{1}{\lambda_i} + \sum_{k < i \leq d} \lambda_i \right).$$

*Proof of Lemma F.4.* According to the definition of Bias in (F.5), using that $\mathbf{w}^* \sim \mathcal{N}(\mathbf{0}, \omega^2 \cdot \mathbf{I}_d)$ we have

$$\mathbb{E}_{\mathbf{w}^*}[\mathrm{Bias}] = \mathbb{E}_{\mathbf{w}^* \sim \mathcal{N}(\mathbf{0}, \omega^2 \cdot \mathbf{I}_d)} \left[ \left\| \mathbf{H}^{\frac{1}{2}} \left( \mathbf{I}_d - \mathbf{X}^\top \mathbf{G}^{-1} \mathbf{X} \right) \mathbf{w}^* \right\|_2^2 \right]$$

$$= \omega^2 \cdot \mathrm{Tr} \left( \mathbf{H} \left( \mathbf{I}_d - \mathbf{X}^\top \mathbf{G}^{-1} \mathbf{X} \right)^2 \right)$$

$$\leq \omega^2 \cdot \mathrm{Tr} \left( \mathbf{H} \left( \mathbf{I}_d - \mathbf{X}^\top \left( \frac{n}{\eta} \cdot \left( \frac{2}{t} + \frac{\sigma^2}{1 - \sigma^2} \left( 1 + \frac{2}{t} \right) \right) \cdot \mathbf{I}_n + \mathbf{A} \right)^{-1} \mathbf{X} \right)^2 \right),$$

where the last inequality follows from Lemma F.3. Notice that the quantity of trace on the right hand side actually corresponds to the bias error of the standard ridge regression with regularization coefficient $\widetilde{\lambda}_{\mathrm{effect}}$ of

$$\widetilde{\lambda}_{\mathrm{effect}}^{\mathrm{Bias}} := \frac{n}{\eta} \cdot \left( \frac{2}{t} + \frac{\sigma^2}{1 - \sigma^2} \left( 1 + \frac{2}{t} \right) \right).$$

Thus by invoking Theorem 1 of (Tsigler & Bartlett, 2023), we can then obtain the result in Lemma F.4. $\square$

**Lemma F.5** (Variance error). *Under Assumption F.1, taking the step size $\eta \lesssim \mathrm{Tr}(\mathbf{H})^{-1}$ and for any $k \in [d]$, with probability at least $1 - 1/\mathrm{poly}(n)$, it holds that*

$$\mathbb{E}_{\boldsymbol{\epsilon}}[\mathrm{Variance}] \lesssim \sigma_\epsilon^2 \cdot \left( \frac{k}{n} + n \cdot \left( \frac{\sigma^2 n}{(1 - \sigma^2)\eta} + \sum_{k < i \leq d} \lambda_i \right)^{-2} \cdot \sum_{k < i \leq d} \lambda_i^2 \right).$$

*Proof of Lemma F.5.* According to the definition of Bias in (F.5), using that $\epsilon_i \sim \mathcal{N}(0, \sigma_\epsilon^2)$ we have

$$\mathbb{E}_{\boldsymbol{\epsilon}}[\mathrm{Variance}] = \mathbb{E}_{\boldsymbol{\epsilon} \sim \mathcal{N}(\mathbf{0}, \sigma_\epsilon^2 \cdot \mathbf{I}_d)} \left[ \left\| \mathbf{H}^{\frac{1}{2}} \mathbf{X}^\top \mathbf{G}^{-1} \boldsymbol{\epsilon} \right\|_2^2 \right]$$

$$= \sigma_\epsilon^2 \cdot \mathrm{Tr} \left( \mathbf{X} \mathbf{H} \mathbf{X}^\top \mathbf{G}^{-2} \right)$$

$$\leq \sigma_\epsilon^2 \cdot \mathrm{Tr} \left( \mathbf{X} \mathbf{H} \mathbf{X}^\top \left( \frac{\sigma^2 n}{(1 - \sigma^2)\eta} \cdot \mathbf{I}_n + \mathbf{A} \right)^{-2} \right)$$

Similar to the proof of Lemma F.4, the above quantity on the right hand side actually corresponds to the variance error of standard ridge regression with regularization coefficient $\widetilde{\lambda}_{\mathrm{effect}}$ of

$$\widetilde{\lambda}_{\mathrm{effect}}^{\mathrm{Var}} := \frac{\sigma^2 n}{(1 - \sigma^2)\eta}.$$

Consequently, by Theorem 1 of (Tsigler & Bartlett, 2023), we can obtain the result in Lemma F.5. $\square$

**Lemma F.6** (Fluctuation error). *Suppose that we choose $\sigma^2 < 1/(d+1)$ and the step size $\eta \lesssim \mathrm{Tr}(\mathbf{H})^{-1}$. Then there exists an event with probability $1 - 1/\mathrm{poly}(n)$ over the randomness of $\mathbf{X}$ on which it holds that*

$$\mathbb{E}_{\mathbf{w}^*, \boldsymbol{\xi}, \boldsymbol{\epsilon}}[\text{Fluctuation}] \lesssim \frac{(\eta \sigma^{-2} \sigma_\epsilon^2 d \cdot \mathrm{Tr}(\mathbf{H})/n + \omega^2) \cdot \|\mathbf{H}\|_2}{N}$$

*Proof of Lemma F.6.* In the proof, we replace the notation $\Delta^{(j)}$ with $\Delta_t^j$ to emphasize the dependence on the reasoning step. From the characterization in Lemma F.2, we have for each path and its expectation over $\boldsymbol{\xi}$, it holds that

$$\begin{aligned}
\mathbf{w}_{t+1}^{(j)} &= (\mathbf{I} - \boldsymbol{\xi}_{t+1}^{(j)} \boldsymbol{\xi}_{t+1}^{(j)\top})(\mathbf{I} - \eta\boldsymbol{\Sigma})(\mathbf{w}_t^{(j)} + \eta\mathbf{X}^\top\mathbf{y}/n) \\
&= (1 - \sigma^2) \cdot (\mathbf{I} - \eta\boldsymbol{\Sigma})(\mathbf{w}_t^{(j)} + \eta\mathbf{X}^\top\mathbf{y}/n) + \sigma^2 \cdot \left(\mathbf{I} - \sigma^{-2}\boldsymbol{\xi}_{t+1}^{(j)}\boldsymbol{\xi}_{t+1}^{(j)\top}\right)(\mathbf{w}_t^{(j)} + \eta\mathbf{X}^\top\mathbf{y}/n)
\end{aligned}$$
$$\mathbf{w}_{t+1} = (1 - \sigma^2)(\mathbf{I} - \eta\boldsymbol{\Sigma})(\mathbf{w}_t + \eta\mathbf{X}^\top\mathbf{y}/n). \tag{F.13}$$

Since there exists an event with probability $1 - 1/\mathrm{poly}(n)$ on which $\mathrm{Tr}(\boldsymbol{\Sigma}) \gtrsim \mathrm{Tr}(\mathbf{H})$, we have that $\eta < 1/\mathrm{Tr}(\boldsymbol{\Sigma})$ with high probability. In order to control the fluctuation error, we begin with deriving a deterministic upper bound on $\mathbf{w}_t$.

**Bounding the expected path.** By (F.13), the quantity $\mathbf{g}_t = \mathbf{w}_t + \eta\mathbf{X}^\top\mathbf{y}$ can be iteratively characterized as follows:

$$\begin{aligned}
\mathbf{g}_{t+1} &= (1 - \sigma^2)(\mathbf{I} - \eta\boldsymbol{\Sigma})\mathbf{g}_t + \eta\mathbf{X}^\top\mathbf{y}/n \\
&= \sum_{k=0}^t (1 - \sigma^2)^k (\mathbf{I} - \eta\boldsymbol{\Sigma})^k \eta\mathbf{X}^\top\mathbf{y}/n \\
&= \sum_{k=0}^t (\mathbf{I} - \sigma^2\mathbf{I} - \eta\boldsymbol{\Sigma} + \eta\sigma^2\boldsymbol{\Sigma})^k \eta\boldsymbol{\Sigma}\mathbf{w}^* \\
&\quad + \sum_{k=0}^t (\mathbf{I} - \sigma^2\mathbf{I} - \eta\boldsymbol{\Sigma} + \eta\sigma^2\boldsymbol{\Sigma})^k \eta\mathbf{X}^\top\boldsymbol{\epsilon}/n,
\end{aligned}$$

To this end, we define $p(z) = \sum_{k=0}^t (1 - \sigma^2 - z + \sigma^2 z)^k$. We can bound the scalar polynomials $p(z)$, $p(z) \cdot z$ and $p^2(z) \cdot z$ on $[0, 1)$ as

$$\begin{aligned}
p(z) &\leq \frac{1}{\sigma^2 + (1 - \sigma^2)z}; \\
p(z) \cdot z &\leq \frac{z}{\sigma^2 + (1 - \sigma^2)z} \lesssim (\sigma^{-2}z) \wedge 1; \tag{F.14} \\
p^2(z) \cdot z &\leq \frac{z}{\left(\sigma^2 + (1 - \sigma^2)z\right)^2} \lesssim (\sigma^{-4}z) \wedge z^{-1}. \tag{F.15}
\end{aligned}$$

We begin with the first term. It follows from (F.14) that

$$\|p(\eta\boldsymbol{\Sigma}) \cdot \eta\boldsymbol{\Sigma}\|_2 \lesssim (\sigma^{-2} \cdot \eta\|\boldsymbol{\Sigma}\|_2) \wedge 1.$$

Therefore the first term can be upper bounded by $\left((\sigma^{-2}\eta\|\boldsymbol{\Sigma}\|_2) \wedge 1\right) \cdot \|\mathbf{w}^*\|_2$. For the second term, we have that

$$\mathbb{E}_{\boldsymbol{\epsilon}}\left[\left\|\sum_{k=0}^t (\mathbf{I} - \sigma^2\mathbf{I} - \eta\boldsymbol{\Sigma} + \eta\sigma^2\boldsymbol{\Sigma})^k \eta\mathbf{X}^\top\boldsymbol{\epsilon}/n\right\|_2^2\right] = \frac{\eta\sigma_\epsilon^2}{n} \cdot \mathrm{Tr}\left(p(\eta\boldsymbol{\Sigma}) \cdot \eta\boldsymbol{\Sigma} \cdot p(\eta\boldsymbol{\Sigma})\right),$$

And therefore we have by (F.15) that

$$\begin{aligned}
\mathbb{E}_{\boldsymbol{\epsilon}, \mathbf{w}^*}[\sup_{t \geq 0} \|\mathbf{g}_t\|_2^2] &\lesssim \frac{\eta\sigma_\epsilon^2}{n} \cdot \sigma^{-4}\mathrm{Tr}(\boldsymbol{\Sigma}) + \left(1 \wedge \sigma^{-2}\eta\|\boldsymbol{\Sigma}\|_2\right)\|\mathbf{w}^*\|_2^2 \\
&\lesssim \frac{\eta\sigma_\epsilon^2}{n} \sigma^{-4}\mathrm{Tr}(\boldsymbol{\Sigma}) + \|\mathbf{w}^\star\|_2^2.
\end{aligned}$$

**Bounding the fluctuation.** In the following, we use $\mathbf{\Lambda}_t^{(j)} = (\mathbf{I} - \sigma^{-2}\boldsymbol{\xi}_{t+1}^{(j)}\boldsymbol{\xi}_{t+1}^{(j)\top})$ for abbreviation. The fluctuation term $\Delta_t^{(j)}$ follows that

$$
\begin{aligned}
\Delta_{t+1}^{(j)} &= \mathbf{w}_{t+1}^{(j)} - \mathbf{w}_{t+1} \\
&= (1 - \sigma^2) \cdot (\mathbf{I} - \eta\mathbf{\Sigma}) \cdot \Delta_t^{(j)} + \sigma^2 \cdot \mathbf{\Lambda}_t^{(j)} \cdot (\mathbf{w}_t^{(j)} + \eta\mathbf{X}^\top\mathbf{y}).
\end{aligned}
\tag{F.16}
$$

For each $t$, we have that $\mathbf{\Lambda}_t^{(j)}$ is independent with $\mathbf{w}_t^{(j)}$ and is of zero mean. Consequently we have that $\mathbb{E}[\Delta_t^{(j)}] = 0$ for any $t \geq 0$. Besides, it can be easily verified by induction that $\Delta_t^{(j)}, j \leq N$ are independent and identically distributed. Thanks to this, we have that

$$
\begin{aligned}
\mathbb{E}\left[\left\|N^{-1}\sum_{j\leq N}\Delta_t^{(j)}\right\|_{\mathbf{H}}^2\right] &= \mathbb{E}\left[N^{-2}\sum_{j\leq N}\Delta_t^{(j)\top}\mathbf{H}\Delta_t^{(j)} + N^{-2}\sum_{j<k}\Delta_t^{(j)\top}\mathbf{H}\Delta_t^{(k)}\right] \\
&= N^{-1}\langle\mathbf{H}, \mathbb{E}[\Delta_t^{(j)\top}\Delta_t^{(j)}]\rangle.
\end{aligned}
\tag{F.17}
$$

Therefore, it suffices to upper bound the second moment of the fluctuation along a single reasoning path. For simplicity, let us drop the superscript $(j)$ in the subsequent analysis. We study the iteration of the second moment $\mathbf{S}_t = \mathbb{E}[\Delta_t\Delta_t^\top]$. Rewriting (F.16), we get that

$$
\begin{aligned}
\Delta_{t+1} &= (1 - \sigma^2) \cdot (\mathbf{I} - \eta\mathbf{\Sigma})\Delta_t + \sigma^2\mathbf{\Lambda}_t\Delta_t \\
&\quad + \sigma^2\mathbf{\Lambda}_t \cdot (\mathbf{w}_t + \eta\mathbf{X}^\top\mathbf{y}).
\end{aligned}
$$

Note that $\mathbf{\Lambda}_t$ and $\Delta_t$ are zero mean and independent, we have that

$$
\begin{aligned}
\mathbf{S}_{t+1} &= (1 - \sigma^2)^2 \cdot (\mathbf{I} - \eta\mathbf{\Sigma})\mathbf{S}_t(\mathbf{I} - \eta\mathbf{\Sigma}) \\
&\quad + \sigma^4 \cdot \mathbb{E}[\mathbf{\Lambda}_t\Delta_t\Delta_t^\top\mathbf{\Lambda}_t^\top] + \sigma^4\eta^2 \cdot \mathbb{E}[\mathbf{\Lambda}_t\mathbf{w}_t\mathbf{w}_t^\top\mathbf{\Lambda}_t^\top] \\
&= (1 - \sigma^2)^2 \cdot (\mathbf{I} - \eta\mathbf{\Sigma})\mathbf{S}_t(\mathbf{I} - \eta\mathbf{\Sigma}) \\
&\quad + \sigma^4\big(\mathrm{Tr}(\mathbf{S}_t)\mathbf{I} + \mathrm{diag}(\mathbf{S}_t)\big) + \sigma^4 \cdot \big(\mathrm{Tr}(\mathbf{g}_t\mathbf{g}_t^\top)\mathbf{I} + \mathrm{diag}(\mathbf{g}_t\mathbf{g}_t^\top)\big).
\end{aligned}
$$

Here the second identity follows from Lemma F.7 and $\mathbf{g}_t = \mathbf{w}_t + \eta\mathbf{X}^\top\mathbf{y}$. The structure of this iteration has two folds. The first part is that the gradient step, together with the average effect of the noise term, help to decay the second moment of the fluctuation. The second part is that the noise term re-allocate the fluctuation in the last step to the current step in an isotropic manner. Since $\mathrm{Tr}(\mathbf{A})$ prevails over $\mathrm{diag}(\mathbf{A})$, we can continue as

$$
\begin{aligned}
\mathrm{Tr}(\mathbf{S}_{t+1}) &\leq (1 - \sigma^2)^2 \cdot \|\mathbf{I} - \eta\mathbf{\Sigma}\|_2^2\mathrm{Tr}(\mathbf{S}_t) + \sigma^4(d + 1) \cdot \big(\mathrm{Tr}(\mathbf{S}_t) + \mathrm{Tr}(\mathbf{g}_t\mathbf{g}_t^\top)\big) \\
&\leq \Big((1 - \sigma^2)^2 \cdot \|\mathbf{I} - \eta\mathbf{\Sigma}\|_2^2 + \sigma^4(d + 1)\Big) \cdot \mathrm{Tr}(\mathbf{S}_t) + \sigma^4(d + 1)\max_{t\geq 0}\|\mathbf{g}_t\|_2^2.
\end{aligned}
\tag{F.18}
$$

Based on our assumption that $\sigma^2 < (d + 1)^{-1}$, it holds by the convexity of the quadratic function that

$$
(1 - \sigma^2)^2 \cdot \|\mathbf{I} - \eta\mathbf{\Sigma}\|_2^2 + \sigma^4(d + 1) \leq (1 - \sigma^2)^2 + \sigma^4(d + 1)
$$

$$
\leq 1 - \frac{d\sigma^2}{d + 1}.
$$

Plugging this back to (F.18), we have that

$$
\begin{aligned}
\mathrm{Tr}(\mathbf{S}_{t+1}) &\leq \frac{\sigma^4 \cdot (d + 1) \cdot \max_{t\geq 0}\|\mathbf{g}_t\|_2^2}{1 - (1 - \sigma^2)^2 \cdot \|\mathbf{I} - \eta\mathbf{\Sigma}\|_2^2 - \sigma^4(d + 1)} \\
&\leq \frac{(d + 1)^2\sigma^2}{d} \cdot \max_{t\geq 0}\|\mathbf{g}_t\|_2^2.
\end{aligned}
$$

Now we can leverage (F.17) and get that

$$\mathbb{E}_{\boldsymbol{\epsilon},\mathbf{w}^\star,\boldsymbol{\xi}}\left[\left\|\frac{1}{N}\sum_{j=1}^N \Delta^{(j)}\right\|_{\mathbf{H}}^2\right] \leq \mathbb{E}_{\boldsymbol{\epsilon},\mathbf{w}^\star}\left[N^{-1}\mathrm{Tr}(\mathbf{S}_t)\cdot\|\mathbf{H}\|_2\right]$$

$$\lesssim \frac{(d+1)^2\sigma^2}{Nd}\cdot\left(\frac{\eta\sigma_\epsilon^2}{n}\cdot\sigma^{-4}\mathrm{Tr}(\boldsymbol{\Sigma})+\mathbb{E}_{\mathbf{w}^\star}[\|\mathbf{w}^\star\|_2^2]\right)\cdot\|\mathbf{H}\|_2$$

$$\lesssim \frac{(\eta\sigma^{-2}\sigma_\epsilon^2 d\cdot\mathrm{Tr}(\mathbf{H})/n+\omega^2)\cdot\|\mathbf{H}\|_2}{N}.$$

The last inequality use that $\mathrm{Tr}(\boldsymbol{\Sigma})\lesssim\mathrm{Tr}(\mathbf{H})$ with high probability. This concludes the proof for the fluctuation error. $\quad\square$

Now with the above lemmas, we are ready to conclude and prove Theorem E.2 for Example 4.2.

*Proof of Theorem E.2 for Example 4.2.* Combining Lemma F.2, Lemma F.4, Lemma F.5, and Lemma F.6 gives the desired result. $\quad\square$

## F.4. Proof of Theorem 5.2

### F.4.1. PROOF FOR EXAMPLE 4.1

*Proof of Theorem 5.2 for Example 4.1.* This follows directly from Theorem E.2 for Example 4.1 and the proof of Proposition 5.1. $\quad\square$

### F.4.2. PROOF FOR EXAMPLE 4.2

*Proof of Theorem 5.2 for Example 4.2.* This follows from Theorem E.2 for Example 4.2, and repeating the proof of Proposition 5.2 for $k_{\mathrm{Bias}}^*$ and $k_{\mathrm{Var}}^*$ in Theorem E.2. $\quad\square$

## F.5. Technical Results

**Lemma F.7.** *For any deterministic matrix $\mathbf{A}\in\mathbb{R}^{d\times d}$ and $\boldsymbol{\xi}\sim\mathcal{N}(\mathbf{0}_d,\mathbf{I}_d)$, it holds that*

$$\mathbb{E}[(\mathbf{I}-\boldsymbol{\xi}\boldsymbol{\xi}^\top)\mathbf{A}(\mathbf{I}-\boldsymbol{\xi}\boldsymbol{\xi}^\top)]=\mathrm{Tr}(\mathbf{A})\mathbf{I}_d+\mathrm{diag}(\mathbf{A}),$$

*where $(\mathrm{diag}(\mathbf{A}))_{ij}=\delta_{ij}\cdot A_{ij}$ and $\delta_{ij}$ is the Kronecker delta.*

*Proof of Lemma F.7.* Note that the $(i,j)$-entry of $\mathbf{I}-\boldsymbol{\xi}\boldsymbol{\xi}^\top$ is $\delta_{ij}-\xi_i\xi_j$. First of all, it is clear that whenever $|\{i,j\}\setminus\{k,l\}|\geq 1$ or $|\{k,l\}\setminus\{i,j\}|\leq 1$, we have that $\mathbb{E}[(\delta_{ij}-\xi_i\xi_j)\cdot(\delta_{kl}-\xi_k\xi_l)]=0$. So the only non-trivial cases are that: (i) $i=j=k=l$; (ii) $\{i,j\}=\{k,l\}$ and $i\neq j$. For the first case, we have that $\mathbb{E}[(\delta_{ij}-\xi_i\xi_j)\cdot(\delta_{kl}-\xi_k\xi_l)]=\mathbb{E}[(\xi_i\xi_j)^2]=1$. For the second case, we have that $\mathbb{E}[(1-\xi_i^2)^2]=\mathbb{E}[\xi_i^4]-\mathbb{E}[\xi_i^2]^2=2$.

Given this we have for $i\neq j$ that

$$\mathbb{E}[\boldsymbol{\Lambda}\mathbf{A}\boldsymbol{\Lambda}]_{i,j}=\mathbb{E}\left[\sum_{k,l=1}^d\boldsymbol{\Lambda}_{ik}\mathbf{A}_{kl}\boldsymbol{\Lambda}_{lj}\right]=0,$$

because each summand is zero since $i\neq j$. For the diagonal terms, we have that

$$\mathbb{E}[\boldsymbol{\Lambda}\mathbf{A}\boldsymbol{\Lambda}]_{i,i}=\mathbb{E}\left[\sum_{k,l=1}^d\boldsymbol{\Lambda}_{ik}\mathbf{A}_{kl}\boldsymbol{\Lambda}_{li}\right]$$

$$=\mathbb{E}\left[\sum_{k=1}^d\boldsymbol{\Lambda}_{ik}\mathbf{A}_{kk}\boldsymbol{\Lambda}_{ki}\right]$$

$$=\mathbb{E}\left[\sum_{k\neq i}\boldsymbol{\Lambda}_{ik}\mathbf{A}_{kk}\boldsymbol{\Lambda}_{ki}\right]+\mathbb{E}[\boldsymbol{\Lambda}_{ii}\mathbf{A}_{ii}\boldsymbol{\Lambda}_{ii}]$$

$$=\mathrm{Tr}(\mathbf{A})+\mathbf{A}_{ii}.$$

Thus the desired result follows. □

# G. Proofs for Section 6

**Notation** We let $[n]$ denote the set of indices from 1 to $n$. Boldface uppercase letters such as $\mathbf{X}$ represent matrices, while boldface lowercase letters such as $\mathbf{x}$ denote vectors. Specifically, $\mathbf{x}[i]$ denotes the $i$-th element of $\mathbf{x}$.

## G.1. Proof of Theorem 6.1

*Proof of Proposition 6.1.* Considering that we sample $N$ different $\mathbf{w}_t$ from the distribution $\{p(\mathbf{w}_t = \mathbf{w})\}_{\mathbf{w} \in \mathcal{W}}$ to obtain $\mathbf{W} = \{\mathbf{w}_t^{(1)}, \dots, \mathbf{w}_t^{(N)}\}$. Let $\texttt{Count}(\mathbf{w})$ represent the frequency of occurrence of $\mathbf{w}$ in $\mathbf{W}$. For each $\mathbf{w}' \in \mathcal{W} \setminus \{\mathbf{w}^*\}$, we upper bound the probability of $\texttt{Count}(\mathbf{w}') > \texttt{Count}(\mathbf{w}^*)$. To this end, we define $N$ random variables $a_1, \cdots, a_N$ such that $a_i = 1$ if $\mathbf{w}_t^{(i)} = \mathbf{w}^*$, $a_i = -1$ if $\mathbf{w}_t^{(i)} = \mathbf{w}'$, and $a_i = 0$ otherwise. This leads to the following bound,

$$\mathbb{P}(\texttt{Count}(\mathbf{w}') > \texttt{Count}(\mathbf{w}^*) \mid \mathbf{w}_0, \mathcal{D}) \le \mathbb{P}\left(\sum_{i=1}^N a_i \le 0 \mid \mathbf{w}_0, \mathcal{D}\right) \le \exp\left(-(p(\mathbf{w}_t = \mathbf{w}^*) - p(\mathbf{w}_t = \mathbf{w}'))^2 \cdot \frac{N}{2}\right),$$

where the last inequality is due to Hoeffding's inequality. Then

$$\sum_{\mathbf{w}' \in \mathcal{W} \setminus \{\mathbf{w}^*\}} \mathbb{P}(\mathbf{w}_{t,N}^{\texttt{mv}} = \mathbf{w}' \mid \mathbf{w}_0, \mathcal{D}) \le \sum_{\mathbf{w}' \in \mathcal{W} \setminus \{\mathbf{w}^*\}} \mathbb{P}(\texttt{Count}(\mathbf{w}') > \texttt{Count}(\mathbf{w}^*) \mid \mathbf{w}_0, \mathcal{D})$$

$$\le \sum_{\mathbf{w}' \in \mathcal{W} \setminus \{\mathbf{w}^*\}} \exp\left(-\frac{N}{2} \cdot (p(\mathbf{w}^*) - p(\mathbf{w}'))^2\right)$$

$$\le |\mathcal{W} \setminus \{\mathbf{w}^*\}| \cdot \exp\left(-\frac{N}{2} \cdot \Delta_t^2\right),$$

where the final inequality is based on the definition of $\Delta_t = p(\mathbf{w}^*) - \max_{\mathbf{w}' \in \mathcal{W} \setminus \{\mathbf{w}^*\}} p(\mathbf{w}')$. Consequently,

$$\mathbb{P}(\mathbf{w}_{t,N}^{\texttt{mv}} = \mathbf{w}^* \mid \mathbf{w}_0, \mathcal{D}) \ge 1 - \sum_{\mathbf{w}' \in \mathcal{W} \setminus \{\mathbf{w}^*\}} \mathbb{P}(\mathbf{w}_{t,N}^{\texttt{mv}} = \mathbf{w}' \mid \mathbf{w}_0, \mathcal{D}) \ge 1 - |\mathcal{W}| \cdot \exp\left(-\frac{N}{2} \cdot \Delta_t^2\right).$$

This completes the proof of Proposition 6.1. □

## G.2. Proof of Theorem 6.2

Here, we first establish bounds for each element in $\tilde{\mathbf{w}}_t$ in Theorem G.1. Next, in Theorem G.2, we prove $\mathbf{w}_T$ will converge to $\mathbf{w}^*$ for both greedy decoding and majority vote algorithm. Lastly, in Theorem G.3, we demonstrate the convergence rate for greedy decoding as shown in Theorem 6.2.

**Lemma G.1.** *Given* $\tilde{\mathbf{w}}_t = \mathbf{w}_{t-1} - \frac{1}{n}\left(\mathbf{X}\mathbf{X}^\top \mathbf{w}_{t-1} - \mathbf{X}\mathbf{Y}^\top\right)$, *where* $\mathbf{Y} = \mathbf{w}^*\mathbf{X} + \epsilon$, *We define* $\mathcal{E}_1$ *as follows:*

$$\mathcal{E}_1 := \left\{ \begin{array}{l} \mathbf{w}^*[i] + \dfrac{2k + \sigma_\epsilon}{n^{1/4}} \ge \tilde{\mathbf{w}}_t[i] \ge \mathbf{w}^*[i] - \dfrac{2k + \sigma_\epsilon}{n^{1/4}}, \\ \textit{specifically when } \mathbf{w}_{t-1} = \mathbf{w}^*, \mathbf{w}^*[i] + \dfrac{\sigma_\epsilon}{n^{1/4}} \ge \tilde{\mathbf{w}}_t[i] \ge \mathbf{w}^*[i] - \dfrac{\sigma_\epsilon}{n^{1/4}} \end{array} \right\},$$

*then* $\mathcal{E}_1$ *holds with probability at least* $1 - \delta$, *where* $\delta = 2\left(d^2 + 2d\right)e^{-cn^{1/2}}$.

*Proof.*

$$\tilde{\mathbf{w}}_t[i] = \mathbf{w}_{t-1}[i] - \frac{1}{n} \sum_{j \in [n], l \in [d]} (x_{ji}x_{jl}\mathbf{w}_{t-1}[l] - x_{ji}x_{jl}\mathbf{w}^*[l]) + \frac{1}{n} \sum_{j \in [n]} x_{ji}\epsilon_i$$

$$= \mathbf{w}_{t-1}[i] - \frac{1}{n}(\mathbf{w}_{t-1}[i] - \mathbf{w}^*[i]) \underbrace{\sum_{j \in [n]} x_{ji}^2}_{A_i} - \frac{1}{n} \sum_{l \in [d], l \neq i} (\mathbf{w}_{t-1}[l] - \mathbf{w}^*[l]) \underbrace{\sum_{j \in [n]} (x_{ji}x_{jl})}_{B_{il}} + \frac{1}{n} \sum_{j \in [n]} x_{ji}\epsilon_i$$

$$= \mathbf{w}_{t-1}[i] - \frac{1}{n}(\mathbf{w}_{t-1}[i] - \mathbf{w}^*[i]) A_i - \frac{1}{n} \sum_{l \in [d], l \neq i} (\mathbf{w}_{t-1}[l] - \mathbf{w}^*[l]) B_{il} + \frac{1}{n} \sum_{j \in [n]} x_{ji}\epsilon_i.$$

Since $x_{ij} \sim \mathcal{N}(0, 1)$ for any $i, j$, by Lemma 2.7.7 and Bernstein's inequality in (Vershynin, 2018), there exists an absolute constant $c_1$ such that

$$\mathbb{P}\{|\sum_i x_{ji}x_{jl}| \leq t\} \leq 2 \exp\left(-c_1 \min\left(\frac{t^2}{\sum_j ||x_{ji}x_{jl}||_{\psi_i}^2}, \frac{t}{\max_j ||x_{ji}x_{jl}||_{\psi_i}}\right)\right),$$

where $||.||_{\psi_1}$ denotes to the sub-exponential norm. Besides, $||x_{ji}x_{jl}||_{\psi_i} \leq ||x_{ji}||_{\psi_2} \cdot ||x_{jk}||_{\psi_2} \leq C_1^2$, with the last inequality derived from the properties of the Gaussian distribution, where $C_1$ is a constant. Furthermore, we have:

$$\mathbb{P}\{|B_{il}| \leq t_1\} \leq 2 \exp\left(-c_1 \min\left(\frac{t_1^2}{nC_1^4}, \frac{t_1}{C_1^2}\cdot\right)\right)$$

Similarly we have

$$\mathbb{P}\{|\sum_{j \in [n]} x_{ji}\epsilon_i| \leq t_2\} \leq 2 \exp\left(-c_2 \min\left(\frac{t_2^2}{nC_1^4\sigma_\epsilon^2}, \frac{t_2}{C_1^2\sigma_\epsilon}\cdot\right)\right)$$

For $A_i = \sum_{j \in [n]} x_{ji}^2$, since $x_{ji}^2 - 1$ are sub-exponential and mean zero random variables, we can directly apply Bernstein's inequality to obtain:

$$\mathbb{P}\{|A_i - n| \leq t_3\} \leq 2 \exp\left(-c_3 \min\left(\frac{t_3^2}{nC_3^4}, \frac{t_3}{C_3^2}\right)\right)$$

By setting $t_1 = t_3 = n^{3/4}, t_2 = \sigma_\epsilon n^{3/4}, c = \frac{\min(c_1, c_2, c_3)}{\max(C_1^4, C_2^4, C_3^4, C_1^2, C_2^2, C_3^2)}$, and applying the derived Theorem G.2, Theorem G.2, Theorem G.2 for all $i, l \in [d]$, we establish that

$$|B_{il}| \leq n^{3/4} \qquad \forall i, l \in [d];$$
$$|\sum_{j \in [n]} x_{ji}\epsilon_i| \leq \sigma_\epsilon n^{3/4} \qquad \forall i \in [d];$$
$$|A_i - n| \leq n^{3/4} \qquad \forall i \in [d],$$

holds with a probability of at least $1 - 2(d^2 + 2d)e^{-cn^{1/2}}$. Hereafter, we condition on Theorem G.2.

By combining Theorem G.2 with Theorem G.2, the following equation is obtained:

$$\tilde{\mathbf{w}}_t[i] = \mathbf{w}_{t-1}[i] - \frac{1}{n}(\mathbf{w}_{t-1}[i] - \mathbf{w}^*[i]) A_i - \frac{1}{n} \sum_{l \in [d], l \neq i} (\mathbf{w}_{t-1}[l] - \mathbf{w}^*[l]) B_{il} + \frac{1}{n} \sum_{j \in [n]} x_{ji}\epsilon_i$$

$$\leq \mathbf{w}^*[i] + \frac{1}{n^{1/4}} \sum_{l \in [d]} |\mathbf{w}_{t-1}[l] - \mathbf{w}^*[l]| + \frac{\sigma_\epsilon}{n^{1/4}}$$

$$\leq \mathbf{w}^*[i] + \frac{2k + \sigma_\epsilon}{n^{1/4}},$$

the final inequality is by $||\mathbf{w}_t||_0 = k \ (t \geq 1)$ and $||\mathbf{w}_0||_0 = 0$. Similarly we have.

$$\tilde{\mathbf{w}}_t[i] \geq \mathbf{w}^*[i] - \frac{1}{n^{1/4}} \sum_{l \in [d]} |\mathbf{w}_{t-1}[l] - \mathbf{w}^*[l]| - \frac{\sigma_\epsilon}{n^{1/4}}$$

$$\geq \mathbf{w}^*[i] - \frac{2k + \sigma_\epsilon}{n^{1/4}}$$

Specifically, when $\mathbf{w}_{t-1} = \mathbf{w}^*$,

$$\mathbf{w}^*[i] + \frac{\sigma_\epsilon}{n^{1/4}} \geq \tilde{\mathbf{w}}_t[i] \geq \mathbf{w}^*[i] - \frac{\sigma_\epsilon}{n^{1/4}}.$$

$\square$

Without loss of generality, in the following we assume the first $k$ elements of $\mathbf{w}^*$ are 1, and others are 0. We define $\mathcal{C}^{(m)}$ as the set of all possible permutations for $[m]$.

**Lemma G.2** (Perfect Accuracy for Both Greedy Decoding and Majority Vote). *Given* $\tilde{\mathbf{w}}_t = \mathbf{w}_{t-1} - \frac{1}{n}\left(\mathbf{XX}^\top \mathbf{w}_{t-1} - \mathbf{XY}^\top\right)$, *where* $\mathbf{Y} = \mathbf{w}^*\mathbf{X} + \epsilon$, *suppose* $\mathcal{E}_1$ *holds,* $\frac{2k+\sigma_\epsilon}{n^{1/4}} < \frac{1}{3}$ *and sampling number* $N$ *is sufficient large, then for all* $t \geq 1$, *we have*

$$\mathbf{w}_t^{\text{maj}\cdot N} = \mathbf{w}_t^{\text{greedy}} = \mathbf{w}^*.$$

*Proof.* Given that $\mathcal{E}_1$ holds, for $t \geq 1$:

$$\begin{cases} \tilde{\mathbf{w}}_t[i] \geq 1 - \frac{2k+\sigma_\epsilon}{n^{1/4}} > 1/2 & i \leq k \\ \tilde{\mathbf{w}}_t[i] \leq \frac{2k+\sigma_\epsilon}{n^{1/4}} < 1/2 & k < i \leq d \end{cases}.$$

In this case we observe that $\tilde{\mathbf{w}}_t[i] > \tilde{\mathbf{w}}_t[j]$ for all $i \leq k$ and $k < i \leq d$. Without loss of generality, we further assume

$$\tilde{\mathbf{w}}_t[1] \geq \tilde{\mathbf{w}}_t[2] \geq \cdots \geq \tilde{\mathbf{w}}_t[k] > \tilde{\mathbf{w}}_t[k+1] \geq \tilde{\mathbf{w}}_t[k+2] \geq \cdots \geq \tilde{\mathbf{w}}_t[d].$$

For $p_{\tilde{\mathbf{w}}_t}[i] = \frac{\max(0, \tilde{\mathbf{w}}_t)}{\sum_{j=1}^d \max(0, \tilde{\mathbf{w}}_t)}$, we also have

$$p_{\tilde{\mathbf{w}}_t}[1] \geq p_{\tilde{\mathbf{w}}_t}[2] \geq \cdots \geq p_{\tilde{\mathbf{w}}_t}[k] > p_{\tilde{\mathbf{w}}_t}[k+1] \geq p_{\tilde{\mathbf{w}}_t}[k+2] \geq \cdots \geq p_{\tilde{\mathbf{w}}_t}[d].$$

Then for $\mathbf{w}' \in \mathcal{W}_{/\mathbf{w}^*}$ where the index of nonzero elements are $e_1, e_2, \ldots, e_k$ (in increasing order), we have

$$\mathbb{P}\left(\mathbf{w}_t = \mathbf{w}^*|\mathbf{w}_{t-1}\right) - \mathbb{P}\left(\mathbf{w}_1 = \mathbf{w}'|\mathbf{w}_{t-1}\right)$$

$$= \sum_{(i_1,\ldots,i_k) \in \mathcal{C}^{(k)}} \left( p_{\tilde{\mathbf{w}}_t}[i_1] \cdot \frac{p_{\tilde{\mathbf{w}}_t}[i_2]}{1 - p_{\tilde{\mathbf{w}}_t}[i_1]} \cdots \frac{p_{\tilde{\mathbf{w}}_t}[i_k]}{1 - \sum_{j<k} p_{\tilde{\mathbf{w}}_t}[i_j]} - p_{\tilde{\mathbf{w}}_t}[e_{i_1}] \cdot \frac{p_{\tilde{\mathbf{w}}_t}[e_{i_2}]}{1 - p_{\tilde{\mathbf{w}}_t}[e_{i_1}]} \cdots \frac{p_{\tilde{\mathbf{w}}_t}[e_{i_k}]}{1 - \sum_{j<k} p_{\tilde{\mathbf{w}}_t}[e_{i_j}]} \right)$$

$$> 0,$$

the last inequality holds because $p_{\tilde{\mathbf{w}}_t}[i] \geq p_{\tilde{\mathbf{w}}_t}[e_i]$ for all $i < k$ and $p_{\tilde{\mathbf{w}}_t}[k] > p_{\tilde{\mathbf{w}}_t}[e_k]$, thus for $t \geq 1$:

$$\mathbb{P}\left(\mathbf{w}_t = \mathbf{w}^*|\mathbf{w}_{t-1}\right) > \mathbb{P}\left(\mathbf{w}_t = \mathbf{w}'|\mathbf{w}_{t-1}\right) \ \forall \mathbf{w}' \in \mathcal{W}_{/\mathbf{w}^*}, \mathbf{w}_{t-1} \in \mathcal{W},$$

Since greedy decoding selects the $\mathbf{w}$ with highest probability, $\mathbf{w}_t^{\text{greedy}} = \mathbf{w}^*$ for all $t \geq 1$. Additionally,

$$\mathbb{P}\left(\mathbf{w}_t = \mathbf{w}^*|\mathbf{w}_0\right) = \sum_{\mathbf{w} \in \mathcal{W}} \mathbb{P}\left(\mathbf{w}_t = \mathbf{w}^*|\mathbf{w}_{t-1} = \mathbf{w}\right) \mathbb{P}\left(\mathbf{w}_{t-1} = \mathbf{w}|\mathbf{w}_0\right)$$

$$> \sum_{\mathbf{w} \in \mathcal{W}} \mathbb{P}\left(\mathbf{w}_t = \mathbf{w}'|\mathbf{w}_{t-1} = \mathbf{w}\right) \mathbb{P}\left(\mathbf{w}_{t-1} = \mathbf{w}|\mathbf{w}_0\right)$$

$$= \mathbb{P}\left(\mathbf{w}_t = \mathbf{w}'|\mathbf{w}_0\right).$$

This implies $\mathbb{P}\left(\mathbf{w}_t = \mathbf{w}^*|\mathbf{w}_0\right) > \mathbb{P}\left(\mathbf{w}_t = \mathbf{w}'|\mathbf{w}_0\right)$ for all $\mathbf{w} \in \mathcal{W}_{/\mathbf{w}^*}$, and according to Theorem 6.1, majority vote will choose $\mathbf{w}_{t,N}^{\text{mv}} = \mathbf{w}^*$ with sufficient large sampling number $N$. $\square$

**Lemma G.3** (Convergence Rate for Majority Vote ). *Given* $\tilde{\mathbf{w}}_t = \mathbf{w}_{t-1} - \frac{1}{n}\left(\mathbf{X}\mathbf{X}^\top \mathbf{w}_{t-1} - \mathbf{X}\mathbf{Y}^\top\right)$, *where* $\mathbf{Y} = \mathbf{w}^*\mathbf{X} + \epsilon$, *suppose* $\mathcal{E}_1$ *holds and* $\frac{2k+\sigma_\epsilon}{n^{1/4}} < \frac{1}{3}$ , *then*

$$\mathbb{P}\left(\mathbf{w}_t = \mathbf{w}^* | \mathbf{w}_0\right) - \max_{\mathbf{w}' \in \mathcal{W}_{/\mathbf{w}^*}} \mathbb{P}\left(\mathbf{w}_t = \mathbf{w}' | \mathbf{w}_0\right) \geq \frac{p_{\text{trans}}}{p_{\text{trans}} + 1 - p_{\text{recurr}}}\left(1 - \left(p_{\text{recurr}} - p_{\text{trans}}\right)^{t-1}\right).$$

*Where*

$$p_{\text{trans}} = \left(1 - \frac{2k + \sigma_\epsilon}{n^{1/4} - (2k + \sigma_\epsilon)}\right)\frac{1}{d^k},$$

$$p_{\text{recurr}} = \left(1 - \frac{\sigma_\epsilon}{n^{1/4} - \sigma_\epsilon}\right)\left(\frac{n^{1/4} - \sigma_\epsilon}{n^{1/4} - \sigma_\epsilon + d\sigma_\epsilon}\right)^k.$$

*Proof.* First, when $\mathbf{w}_{t-1} = \mathbf{w}^*$, we have

$$\begin{cases} \tilde{\mathbf{w}}_t[i] \geq 1 - \frac{\sigma_\epsilon}{n^{1/4}} & i \leq k \\ \tilde{\mathbf{w}}_t[i] \leq \frac{\sigma_\epsilon}{n^{1/4}} & k < i \leq d \end{cases}$$

Let $\tau = \frac{\sigma_\epsilon}{n^{1/4}}$. For $p_{\tilde{\mathbf{w}}_t}[i] = \frac{\max(0, \tilde{\mathbf{w}}_t)}{\sum_{j=1}^d \max(0, \tilde{\mathbf{w}}_t)}$ and $i \leq k$:

$$p_{\tilde{\mathbf{w}}_t}[i] \geq \frac{1 - \tau}{k(1 - \tau) + d\tau} = \frac{1}{k}\frac{k(1 - \tau)}{k(1 - \tau) + d\tau}$$

Hence,

$$\mathbb{P}\left(\mathbf{w}_t = \mathbf{w}^* | \mathbf{w}_{t-1} = \mathbf{w}^*\right) = \sum_{(i_1, \ldots, i_k) \in \mathcal{C}^{(k)}}\left(p_{\tilde{\mathbf{w}}_t}[i_1] \cdot \frac{p_{\tilde{\mathbf{w}}_t}[i_2]}{1 - p_{\tilde{\mathbf{w}}_t}[i_1]} \cdots \frac{p_{\tilde{\mathbf{w}}_t}[i_k]}{1 - \sum_{j < k} p_{\tilde{\mathbf{w}}_t}[i_j]}\right)$$

$$\geq \frac{\left(\frac{1}{k} - \frac{d\tau}{(k(1-\tau)+d\tau)k}\right)^k k!}{\prod_{m=1}^{k-1}\left(1 - m\left(\frac{1}{k} - \frac{d\tau}{(k(1-\tau)+d\tau)k}\right)\right)}$$

$$\geq \left(\frac{1 - \tau}{1 + (d - 1)\tau}\right)^k$$

the last inequality is by let $v = \frac{k(1-\tau)}{k(1-\tau)+d\tau}$

$$\frac{\left(\frac{v}{k}\right)^k k!}{\prod_{m=1}^{k-1}\left(1 - m\frac{v}{k}\right)} \geq \frac{v^k \left(\frac{1}{k}\right)^k k!}{\prod_{m=1}^{k-1}\left((k - (k-1)v)\left(1 - m\frac{1}{k}\right)\right)}$$

$$= \frac{v^k}{(k - (k-1)v)^{k-1}}\frac{\left(\frac{1}{k}\right)^k k!}{\prod_{m=1}^{k-1}\left(1 - m\frac{1}{k}\right)}$$

$$\geq \left(\frac{v}{k - (k-1)v}\right)^k = \left(\frac{1 - \tau}{1 - \tau + d\tau}\right)^k$$

Next, for $\mathbf{w}' \in \mathcal{W}_{/\mathbf{w}^*}$ where the index of nonzero elements are $e_1, e_2, \ldots, e_k$ (increasing order), we have:

$$\mathbb{P}\left(\mathbf{w}_t = \mathbf{w}^* | \mathbf{w}_{t-1}\right) - \mathbb{P}\left(\mathbf{w}_1 = \mathbf{w}' | \mathbf{w}_{t-1}\right)$$

$$= \sum_{(i_1,\ldots,i_k) \in \mathcal{C}^{(k)}} \left( p_{\tilde{\mathbf{w}}_t}[i_1] \cdot \frac{p_{\tilde{\mathbf{w}}_t}[i_2]}{1 - p_{\tilde{\mathbf{w}}_t}[i_1]} \cdots \frac{p_{\tilde{\mathbf{w}}_t}[i_k]}{1 - \sum_{j<k} p_{\tilde{\mathbf{w}}_t}[i_j]} - p_{\tilde{\mathbf{w}}_t}[e_{i_1}] \cdot \frac{p_{\tilde{\mathbf{w}}_t}[e_{i_2}]}{1 - p_{\tilde{\mathbf{w}}_t}[e_{i_1}]} \cdots \frac{p_{\tilde{\mathbf{w}}_t}[e_{i_k}]}{1 - \sum_{j<k} p_{\tilde{\mathbf{w}}_t}[e_{i_j}]} \right)$$

$$> \left( \prod_{i=1}^{k} p_{\tilde{\mathbf{w}}_t}[i] - \prod_{i=1}^{k} p_{\tilde{\mathbf{w}}_t}[e_i] \right) \sum_{(i_1,\ldots,i_k) \in \mathcal{C}^{(k)}} \left( \frac{1}{1 - p_{\tilde{\mathbf{w}}_t}[i_1]} \cdots \frac{1}{1 - \sum_{j<k} p_{\tilde{\mathbf{w}}_t}[i_j]} \right)$$

$$> \left( 1 - \frac{p_{\tilde{\mathbf{w}}_t}[e_i]}{p_{\tilde{\mathbf{w}}_t}[i]} \right) \prod_{i=1}^{k} p_{\tilde{\mathbf{w}}_t}[i] \sum_{(i_1,\ldots,i_k) \in \mathcal{C}^{(k)}} \left( \frac{1}{1 - p_{\tilde{\mathbf{w}}_t}[i_1]} \cdots \frac{1}{1 - \sum_{j<k} p_{\tilde{\mathbf{w}}_t}[i_j]} \right)$$

$$= \left( 1 - \frac{p_{\tilde{\mathbf{w}}_t}[e_i]}{p_{\tilde{\mathbf{w}}_t}[i]} \right) \mathbb{P}\left(\mathbf{w}_t = \mathbf{w}^* | \mathbf{w}_{t-1}\right)$$

Given that $\mathbb{P}\left(\mathbf{w}_t = \mathbf{w}^* | \mathbf{w}_{t-1}\right) > \mathbb{P}\left(\mathbf{w}_t = \mathbf{w}' | \mathbf{w}_{t-1}\right)$ for $\mathbf{w}' \in \mathcal{W}_{/\mathbf{w}^*}$, we have $\mathbb{P}\left(\mathbf{w}_t = \mathbf{w}^* | \mathbf{w}_{t-1}\right) > \frac{1}{|\mathcal{W}|} \geq \frac{1}{d^k}$, when $\mathcal{E}_1$ holds:

$$\mathbb{P}\left(\mathbf{w}_t = \mathbf{w}^* | \mathbf{w}_{t-1}\right) - \mathbb{P}\left(\mathbf{w}_1 = \mathbf{w}' | \mathbf{w}_{t-1}\right) > \left( 1 - \frac{\frac{2k+\sigma_\epsilon}{n^{1/4}}}{1 - \frac{2k+\sigma_\epsilon}{n^{1/4}}} \right) \frac{1}{d^k}$$

Specifically,

$$\mathbb{P}\left(\mathbf{w}_t = \mathbf{w}^* | \mathbf{w}^*\right) - \mathbb{P}\left(\mathbf{w}_1 = \mathbf{w}' | \mathbf{w}^*\right) > \left( 1 - \frac{\frac{\sigma_\epsilon}{n^{1/4}}}{1 - \frac{\sigma_\epsilon}{n^{1/4}}} \right) \mathbb{P}\left(\mathbf{w}_t = \mathbf{w}^* | \mathbf{w}^*\right)$$

Therefore,

$$\mathbb{P}\left(\mathbf{w}_t = \mathbf{w}^* | \mathbf{w}_0\right) - \mathbb{P}\left(\mathbf{w}_t = \mathbf{w}' | \mathbf{w}_0\right)$$

$$= \sum_{\mathbf{w} \in \mathcal{W}} \left( \mathbb{P}\left(\mathbf{w}_t = \mathbf{w}^* | \mathbf{w}_{t-1} = \mathbf{w}\right) - \mathbb{P}\left(\mathbf{w}_t = \mathbf{w}' | \mathbf{w}_{t-1} = \mathbf{w}\right) \right) \mathbb{P}\left(\mathbf{w}_{t-1} = \mathbf{w} | \mathbf{w}_0\right)$$

$$> \sum_{\mathbf{w} \in \mathcal{W}_{/\mathbf{w}^*}} \left( 1 - \frac{2k+\sigma_\epsilon}{n^{1/4} - (2k+\sigma_\epsilon)} \right) \mathbb{P}\left(\mathbf{w}_t = \mathbf{w}^* | \mathbf{w}_{t-1} = \mathbf{w}\right) \mathbb{P}\left(\mathbf{w}_{t-1} = \mathbf{w} | \mathbf{w}_0\right)$$

$$+ \left( 1 - \frac{\sigma_\epsilon}{n^{1/4} - \sigma_\epsilon} \right) \mathbb{P}\left(\mathbf{w}_t = \mathbf{w}^* | \mathbf{w}^*\right) \mathbb{P}\left(\mathbf{w}_{t-1} = \mathbf{w}^* | \mathbf{w}_0\right)$$

$$> \left( 1 - \frac{2k+\sigma_\epsilon}{n^{1/4} - (2k+\sigma_\epsilon)} \right) \frac{1}{d^k} \sum_{\mathbf{w} \in \mathcal{W}_{/\mathbf{w}^*}} \mathbb{P}\left(\mathbf{w}_{t-1} = \mathbf{w} | \mathbf{w}_0\right) + \left( 1 - \frac{\sigma_\epsilon}{n^{1/4} - \sigma_\epsilon} \right) \left( \frac{1-\tau}{1-\tau+d\tau} \right)^k \mathbb{P}\left(\mathbf{w}_{t-1} = \mathbf{w}^* | \mathbf{w}_0\right)$$

$$= \underbrace{\left( 1 - \frac{2k+\sigma_\epsilon}{n^{1/4} - (2k+\sigma_\epsilon)} \right) \frac{1}{d^k}}_{p_{\text{trans}}} (1 - \mathbb{P}\left(\mathbf{w}_{t-1} = \mathbf{w}^* | \mathbf{w}_0\right)) + \underbrace{\left( 1 - \frac{\sigma_\epsilon}{n^{1/4} - \sigma_\epsilon} \right) \left( \frac{n^{1/4} - \sigma_\epsilon}{n^{1/4} - \sigma_\epsilon + d\sigma_\epsilon} \right)^k}_{p_{\text{recurr}}} \mathbb{P}\left(\mathbf{w}_{t-1} = \mathbf{w}^* | \mathbf{w}_0\right)$$

$$> (p_{\text{recurr}} - p_{\text{trans}})^{t-1} \left( \mathbb{P}\left(\mathbf{w}_1 = \mathbf{w}^* | \mathbf{w}_0\right) - \frac{p_{\text{trans}}}{p_{\text{trans}} + 1 - p_{\text{recurr}}} \right) + \frac{p_{\text{trans}}}{p_{\text{trans}} + 1 - p_{\text{recurr}}}$$

$$> \frac{p_{\text{trans}}}{p_{\text{trans}} + 1 - p_{\text{recurr}}} \left( 1 - (p_{\text{recurr}} - p_{\text{trans}})^{t-1} \right)$$

$\square$

### G.3. Proof of Theorem 6.3

To prove Theorem 6.3, we first demonstrate that the majority vote algorithm can achieve perfect accuracy with a high probability given a sufficient large sampling number $N$ (by combining Theorem G.4 and Theorem G.5). Subsequently, for the greedy decoding algorithm, we prove that with high probability, $\mathbf{w}_t^{\text{greedy}}$ will transition between states $\mathbf{w}'$ and $\mathbf{w}''$, where $\mathbf{w}', \mathbf{w}'' \neq \mathbf{w}^*$.

In the following, as we consider the case where $k = 1$, we define $\mathbb{1}_i = [0, \ldots, \underset{i\text{-th}}{1}, 0, \ldots]$ be a vector with a value of 1 at the $i$-th element and 0 elsewhere. Without loss of generality, we assume $\mathbf{w}^* = \mathbb{1}_1$.

**Lemma G.4.** *Consider the case where $n = k = 1, \sigma_\epsilon = 0$, and denote the in-context example as $(\mathbf{x}, \mathbf{w}^\top \mathbf{x})$. Then:*

$$\mathbb{P}\left(\mathbf{w}_{t+2} = \mathbf{w}^* | \mathbf{w}_t = \mathbf{w}\right) > 0$$

*Holds for all $\mathbf{w} \in \mathcal{W}$ with probability at least $1 - \frac{1}{2^{d-1}}$.*

*Proof.*

$$\mathbb{P}\left(\mathbf{w}_{t+2} = \mathbf{w}^* | \mathbf{w}_t = \mathbf{w}\right) = \sum_{\mathbf{w}' \in \mathcal{W}} \mathbb{P}\left(\mathbf{w}_{t+2} = \mathbf{w}^* | \mathbf{w}_{t-1} = \mathbf{w}'\right) \mathbb{P}\left(\mathbf{w}_{t+1} = \mathbf{w}' | \mathbf{w}_t = \mathbf{w}\right)$$

It suffices to demonstrate the existence of a $\mathbf{w}' \in \mathcal{W}$, such that $\mathbb{P}\left(\mathbf{w}_{t+2} = \mathbf{w}^* | \mathbf{w}_{t-1} = \mathbf{w}'\right) \mathbb{P}\left(\mathbf{w}_{t+1} = \mathbf{w}' | \mathbf{w}_t = \mathbf{w}\right) > 0$.

Without losing generality, we let $x_1 > 0$, $\mathbf{w}_t = \mathbb{1}_l$ and for $\mathbf{x} = [x_1, x_2, \ldots, x_d]$ we let $x_1 > 0, x_2 \geq x_3 \cdots \geq x_d$. We have:

$$\tilde{\mathbf{w}}_{t+1}[i] = \mathbf{w}_t[i] - \sum_{j \in [d]} \left(x_i x_j \left(\mathbf{w}_{t-1}[j] - \mathbf{w}^*[j]\right)\right)$$

$$\begin{cases} \tilde{\mathbf{w}}_{t+1}[i] = x_i \left(x_1 - x_l\right) & \text{if } i \neq l \\ \tilde{\mathbf{w}}_{t+1}[i] = 1 + x_l \left(x_1 - x_l\right) & \text{if } i = l \end{cases}.$$

If $x_1 - x_l > 0$, then $\tilde{\mathbf{w}}_{t+1}[1] > 0$, implying the existence of $\mathbf{w}' = \mathbf{w}^*$, such that:

$$\mathbb{P}\left(\mathbf{w}_{t+2} = \mathbf{w}^* | \mathbf{w}_{t-1} = \mathbf{w}'\right) \mathbb{P}\left(\mathbf{w}_{t+1} = \mathbf{w}' | \mathbf{w}_t = \mathbf{w}\right)$$
$$= \mathbb{P}\left(\mathbf{w}_{t+2} = \mathbf{w}^* | \mathbf{w}_{t-1} = \mathbf{w}*\right) \mathbb{P}\left(\mathbf{w}_{t+1} = \mathbf{w}* | \mathbf{w}_t = \mathbf{w}\right)$$
$$= \frac{x_1 \left(x_1 - x_l\right)}{\sum_{i \in [d]} \max\left(0, \tilde{\mathbf{w}}_{t+1}[i]\right)} > 0$$

If $x_1 - x_l < 0$, we consider the case where $x_d < 0$, which occurs with a probability of at least $1 - \frac{1}{2^{d-1}}$. In this case, we ensure $x_d < 0$ to satisfy $x_d \left(x_1 - x_l\right) > 0$. Subsequently, leveraging the condition $x_1 - x_d > 0$, we can choose $\mathbf{w}' = \mathbb{1}_d$ such that:

$$\mathbb{P}\left(\mathbf{w}_{t+2} = \mathbf{w}^* | \mathbf{w}_{t-1} = \mathbf{w}'\right) \mathbb{P}\left(\mathbf{w}_{t+1} = \mathbf{w}' | \mathbf{w}_t = \mathbf{w}\right)$$
$$\geq \frac{x_d \left(x_1 - x_l\right)}{\sum_{i \in [d]} \max\left(0, \tilde{\mathbf{w}}_{t+1}[i]\right)} \cdot \frac{x_1 \left(x_1 - x_d\right)}{\sum_{i \in [d]} \max\left(0, \tilde{\mathbf{w}}_{t+2}[i]\right)} > 0$$

$\square$

**Lemma G.5.** *Consider the case where $n = k = 1, \sigma_\epsilon = 0$, and denote the in-context example as $(\mathbf{x}, \mathbf{w}^\top \mathbf{x})$. There exists a $\zeta > 0$ such that for reasoning steps $T > \frac{2 \ln 1/2}{\ln 1 - \zeta}$ and sufficient large sampling number $N$, it holds that*

$$\mathbf{w}_{T,N}^{\texttt{mv}} = \mathbf{w}^*,$$

*with probability at least $1 - \frac{1}{2^{d-1}}$.*

*Proof.* Referring to Theorem G.4, with probability at least $1 - \frac{1}{2^{d-1}}$, $\mathbb{P}\left(\mathbf{w}_{t+2} = \mathbf{w}^* | \mathbf{w}_t = \mathbf{w}\right) > 0$ holds for all $\mathbf{w} \in \mathcal{W}$, define

$$\zeta = \min_{\mathbf{w} \in \mathcal{W}} \mathbb{P}\left(\mathbf{w}_{t+2} = \mathbf{w}^* | \mathbf{w}_t = \mathbf{w}\right).$$

Assume $t = 2q + 1$ (if not, since $\mathbb{P}\left(\mathbf{w}_t = \mathbf{w}^* | \mathbf{w}_0\right) \geq \mathbb{P}\left(\mathbf{w}_{t-1} = \mathbf{w}^* | \mathbf{w}_0\right)$, we can set $t - 1 = 2q + 1$)

$$
\begin{aligned}
& \mathbb{P}\left(\mathbf{w}_{2q+1} = \mathbf{w}^* | \mathbf{w}_0\right) \\
= & \sum_{\mathbf{w} \in \mathcal{W}} \mathbb{P}\left(\mathbf{w}_{2q+1} = \mathbf{w}^* | \mathbf{w}_{2q-1} = \mathbf{w}\right) \mathbb{P}\left(\mathbf{w}_{2q-1} = \mathbf{w} | \mathbf{w}_0\right) \\
= & \sum_{\mathbf{w} \in \mathcal{W}_{/\mathbf{w}^*}} \mathbb{P}\left(\mathbf{w}_{2q+1} = \mathbf{w}^* | \mathbf{w}_{2q-1} = \mathbf{w}\right) \mathbb{P}\left(\mathbf{w}_{2q-1} = \mathbf{w} | \mathbf{w}_0\right) + \mathbb{P}\left(\mathbf{w}_{2q+1} = \mathbf{w}^* | \mathbf{w}_{2q-1} = \mathbf{w}^*\right) \mathbb{P}\left(\mathbf{w}_{2q-1} = \mathbf{w}^* | \mathbf{w}_0\right) \\
\geq & \zeta\left(1 - \mathbb{P}\left(\mathbf{w}_{2q-1} = \mathbf{w}^* | \mathbf{w}_0\right)\right) + \mathbb{P}\left(\mathbf{w}_{2q-1} = \mathbf{w}^* | \mathbf{w}_0\right) \\
\geq & (1 - \zeta)^k \left(\mathbb{P}\left(\mathbf{w}_1 = \mathbf{w}^* | \mathbf{w}_0\right) - 1\right) + 1 \geq 1 - (1 - \zeta)^k
\end{aligned}
$$

If $k > \frac{\ln 1/2}{\ln(1-\zeta)}$, then $\mathbb{P}\left(\mathbf{w}_{2q+1} = \mathbf{w}^* | \mathbf{w}_0\right) > 1/2$, and therefore:

$$
\mathbb{P}\left(\mathbf{w}_t = \mathbf{w}^* | \mathbf{w}_0\right) > \frac{1}{2} > 1 - \mathbb{P}\left(\mathbf{w}_t = \mathbf{w}^* | \mathbf{w}_0\right) > \mathbb{P}\left(\mathbf{w}_t = \mathbf{w}' | \mathbf{w}_0\right) \ \forall \mathbf{w}' \in \mathcal{W}_{/\mathbf{w}^*}
$$

In this case, by Theorem 6.1, with sufficient large sample number $N$, $\mathbf{w}_{T,N}^{\texttt{mv}} = \mathbf{w}^*$. $\qquad \square$

**Lemma G.6.** *Consider the case where $n = k = 1, \sigma_\epsilon = 0$, and denote the in-context example as $\left(\mathbf{x}, \mathbf{w}^\top \mathbf{x}\right)$. Then*

$$
\mathbf{w}_t^{\texttt{greedy}} \neq \mathbf{w}^*
$$

*holds with probability at least $1 - \frac{2}{d} - \frac{1}{2^{d-1}}$.*

*Proof.* Here, we directly construct a case where, with a high probability, the greedy decoding will become stuck between two stages and fail to reach the state $\mathbf{w}^*$.

Without loss of generality, we assume $x_1 > 0$, and we select $x_2$ and $x_3$ such that $x_2 = \max_{i>1} x_i$ and $x_3 = \max_{i>1}\left(-x_i\right)$. With a probability of $1 - \sum_{r=1}^{d-1} \frac{1}{r+1} \frac{\binom{d-1}{r}}{2^{d-1}} - \frac{1}{2^{d-1}} > 1 - \frac{2}{d} - \frac{1}{2^{d-1}}$, it holds that $x_2 > x_1 > 0$ and $x_3 < 0$.

In this case,

$$
\tilde{\mathbf{w}}_1[2] = x_1 x_2 > x_1 x_j = \tilde{\mathbf{w}}_1[j],
$$

holds for all $j \in [d], j \neq 2$. Then $\mathbf{w}_1^{\texttt{greedy}} = \mathbf{w}' \neq \mathbf{w}^*$ where $\mathbf{w}' = \mathbb{1}_2$. Similarly,

$$
\begin{cases}
\tilde{\mathbf{w}}_2[i] = x_i\left(x_1 - x_2\right) & \text{if } i \neq 2 \\
\tilde{\mathbf{w}}_2[i] = 1 + x_i\left(x_1 - x_2\right) & \text{if } i = 2
\end{cases}
$$

If $\arg\max_{i \in [d]} \tilde{\mathbf{w}}_2[i] = 2$, then $\mathbf{w}_2^{\texttt{greedy}} = \mathbf{w}'$, thus for $\mathbf{w}_t^{\texttt{greedy}} = \mathbf{w}' \neq \mathbf{w}^*$ holds when $t \geq 1$. . If $\arg\max_{i \in [d]} \tilde{\mathbf{w}}_2[i] \neq 2$, as $x_1 - x_2 < 0$,

$$
\tilde{\mathbf{w}}_2[3] = x_3\left(x_1 - x_2\right) > x_i\left(x_1 - x_2\right) = \tilde{\mathbf{w}}_2[j],
$$

holds for all $j \in [d], j \neq 3$. In this case, we have $\mathbf{w}_2 = \mathbf{w}'' \neq \mathbf{w}^*$ where $\mathbf{w}'' = \mathbb{1}_3$ and for $\tilde{\mathbf{w}}_3$:

$$
\begin{cases}
\tilde{\mathbf{w}}_3[i] = x_i\left(x_1 - x_3\right) & \text{if } i \neq 3 \\
\tilde{\mathbf{w}}_3[i] = 1 + x_i\left(x_1 - x_3\right) & \text{if } i = 3
\end{cases}
$$

Similarly, if $\arg\max_{i \in [d]} \tilde{\mathbf{w}}_3[i] = 3$, then $\mathbf{w}_3^{\texttt{greedy}} = \mathbf{w}''$, thus for $\mathbf{w}_t^{\texttt{greedy}} = \mathbf{w}'' \neq \mathbf{w}^*$ holds when $t \geq 2$.

If $\arg\max_{i \in [d]} \tilde{\mathbf{w}}_2[i] \neq 2$, as $\left(x_1 - x_3\right) > 0$, we know that $\mathbf{w}_3^{\texttt{greedy}} = \mathbf{w}'$, then $\mathbf{w}_4^{\texttt{greedy}} = \mathbf{w}''$, $\mathbf{w}_5^{\texttt{greedy}} = \mathbf{w}'$...

In conclusion, $\mathbf{w}_t^{\texttt{greedy}}$ will be either $\mathbf{w}'$ or $\mathbf{w}''$ for $t > 0$, thus $\mathbf{w}_t^{\texttt{greedy}} \neq \mathbf{w}^*$ for $t > 0$. $\qquad \square$

## H. Prompt Examples

---

**Prompt For GSM8K with Assigned Token Budget**

You are a math problem solver. I will give you a problem from the Grade School Math 8K dataset (GSM8K). At the end, provide the final answer as a single integer.

Example: Problem: There are 15 trees in the grove. Grove workers will plant trees in the grove today. After they are done, there will be 21 trees. How many trees did the grove workers plant today? Answer (You should choose different reasoning method based on different tokens limit):

Case 1 (low token budgets, for example 20): We have token limits 20. The answer is ##6##. [END]

Case 2 (medium token budgets, for example 100): We have token limits 100. 21 - 15 = 6. The answer is ##6##. [END]

Case 3 (high token budgets, for example 200): We have token limits 200. There are 15 trees originally. Then there were 21 trees after some more were planted. So there must have been 21 - 15 = 6. The answer is ##6##. [END]

Case 4 (sufficient token budgets, for example 500): We have token limits 500. There are 15 trees originally. Then there were 21 trees after some more were planted. So there must have been 21 - 15 = 6. [...(more thoughts such as check answer to satisfy tokens limit)] The answer is ##6##. [END]

Important: You should try your best to use around {token_limit} tokens in your reasoning steps.

If you feel like you are finished early, spend the extra tokens trying to double check your work until you are absolutely sure that you have the correct answer.

Here's the problem:

{problem}

Solve this problem, use around {token_limit} tokens in your reasoning, provide the final answer as a single integer, and put your final answer in this format: "The answer is ##your answer##.", and end this chat with '[END]'

---

For the MATH dataset, we simply replaced the "Grade School Math 8K dataset (GSM8K)" (first line in above prompt) with "MATH."

