# OpenReview forum: "Towards Theoretical Understanding of Transformer Test-Time Computing: Investigation on In-Context Linear Regression"
_ICML.cc/2026/Conference — ICML 2026 regular_

### Official Review · Reviewer_LBb7 · 2026-02-20

**Soundness:** 4
**Presentation:** 3
**Significance:** 3
**Originality:** 3
**Overall Recommendation:** 5
**Confidence:** 4

**Summary:**

This work proposes a theoretical framework to understand transformer test-time computation by incorporating randomness and sampling into decoding. It studies how sampling-based strategies such as best-of-N affect performance under both continuous and discrete coefficient scenarios. Theoretical findings are also supported by experimental results.

**Compliance With Llm Reviewing Policy:**

Affirmed.

**Final Justification:**

Main concern has been resolved. I appreciate the team's effort.

**Key Questions For Authors:**

Please refer to the weakness.

**Limitations:**

Yes

**Strengths And Weaknesses:**

## Strength:

1. The theoretical framework in this work incorporates randomness and sampling to simulate LLM decoding, enabling detailed analysis of widely used inference strategies.

2. The theories in this work are supported by extensive experiments demonstrating that the framework can capture trends in real-world LLM performance.

3. The theoretical analysis of the accuracy is potentially valuable for transferable evaluation.

## Weakness:

1. More LLM backbones beyond Llama-3.1 will be appreciated

2. An open-source repository of this work will be appreciated.

---

> ### Author Rebuttal · Authors · 2026-03-31
>
> We sincerely appreciate your thorough review of our work. We value your constructive feedback, and we have addressed your questions in detail below:
>
> > W1: More LLM backbones beyond Llama-3.1 will be appreciated
>
> **R1**: Thanks for your suggestion! We appreciate the suggestion to evaluate our framework on a wider range of architectures. To further demonstrate the generalizability of our theoretical insights, we have extended our experiments to include Qwen2.5-7B-Instruct.
>
> The following tables summarize the prediction accuracy across different configurations. The values represent the average absolute difference between predicted and ground-truth accuracy ($|\hat{acc} - acc|$).
>
> ** Qwen2.5-7B-Instruct (GSM8K)**
> | T   | Tokens (N=1) | N=3    | N=8    | N=16   | Avg    |
> |-----|--------------|--------|--------|--------|--------|
> | T2  | 187.6K       | 0.0052 | 0.0091 | 0.0158 | 0.0100 |
> | T3  | 256.5K       | 0.0058 | 0.0112 | 0.0219 | 0.0130 |
> | T4  | 354.3K       | 0.0040 | 0.0136 | 0.0206 | 0.0127 |
> | T5  | 413.9K       | 0.0053 | 0.0144 | 0.0252 | 0.0150 |
> | T6  | 423.6K       | 0.0070 | 0.0134 | 0.0231 | 0.0145 |
> | T7  | 441.2K       | 0.0042 | 0.0104 | 0.0218 | 0.0121 |
>
> **Qwen2.5-7B-Instruct (MATH Subset)**
> | T   | Tokens (N=1) | N=3    | N=8    | N=16   | Avg    |
> |-----|--------------|--------|--------|--------|--------|
> | T2  | 61.4K        | 0.0019 | 0.0023 | 0.0024 | 0.0022 |
> | T3  | 71.4K        | 0.0064 | 0.0040 | 0.0076 | 0.0060 |
> | T4  | 94.3K        | 0.0043 | 0.0107 | 0.0100 | 0.0083 |
> | T5  | 113.8K       | 0.0091 | 0.0122 | 0.0142 | 0.0118 |
> | T6  | 119.5K       | 0.0045 | 0.0064 | 0.0129 | 0.0079 |
>
> **Llama-3.1-8B-Instruct (GSM8K)**
> | T   | Tokens (N=1) | N=3    | N=8    | N=16   | Avg    |
> |-----|--------------|--------|--------|--------|--------|
> | T2  | 215.6K       | 0.0135 | 0.0037 | 0.0008 | 0.0060 |
> | T3  | 278.7K       | 0.0155 | 0.0080 | 0.0074 | 0.0103 |
> | T4  | 342.1K       | 0.0087 | 0.0071 | 0.0066 | 0.0075 |
> | T5  | 400.1K       | 0.0104 | 0.0143 | 0.0110 | 0.0119 |
>
> **Llama-3.1-8B-Instruct (MATH Subset)**
> | T   | Tokens (N=1) | N=3    | N=8    | N=16   | Avg    |
> |-----|--------------|--------|--------|--------|--------|
> | T2  | 56.9K        | 0.0021 | 0.0078 | 0.0004 | 0.0034 |
> | T3  | 60.0K        | 0.0060 | 0.0114 | 0.0097 | 0.0091 |
> | T4  | 77.9K        | 0.0032 | 0.0193 | 0.0104 | 0.0110 |
> | T5  | 96.4K        | 0.0071 | 0.0159 | 0.0135 | 0.0122 |
> | T6  | 114.1K       | 0.0076 | 0.0231 | 0.0176 | 0.0161 |
>
> As illustrated, our algorithm accurately predicts model performance under computationally expensive settings using only data from low-cost configurations (fewer reasoning tokens $T$ and smaller sampling numbers $N$). These results underscore the practical applicability and robustness of our theoretical framework across diverse LLM backbones. We will add these experiments in our revised version.
>
> > W2: An open-source repository of this work will be appreciated.
>
> **R2**: Thanks for your suggestion, We are currently organizing our code and data processing scripts and plan to release them in a public repository upon the issue of this work.
>
> Please let us know if you have any further questions; we are happy to provide additional clarification.

---

> > ### Author Rebuttal · Reviewer_LBb7 · 2026-04-02
> >
> > Main concern has been resolved. I appreciate the team's effort.

---

### Official Review · Reviewer_162g · 2026-03-09

**Soundness:** 4
**Presentation:** 3
**Significance:** 2
**Originality:** 2
**Overall Recommendation:** 4
**Confidence:** 3

**Summary:**

This paper theoretically studies transformer inference on in-context linear regression tasks.  The paper extends previous theoretical settings that use deterministic transformer outputs, to simulate sampling-based decoding via noise injection.

The problem setup mimics [1] (Huang et al., 2025.): throughout the in-context regression tasks considered, the target is to estimate the ground truth regression parameter (the task is called in-context weight learning). The setting incorporates CoT reasoning, where the intermediate estimates of the parameter mimic the reasoning steps. It is assumed that the transformer implements multi-step gradient descent on the target parameter (deterministic version), and stochasticity is added via noise injection. The theory considers a one-layer linear attention Transformer. The precise contributions of the paper are:
- A theoretical framework that models transformer inference better through the stochasticity implemented by the noise injection. More concretely, they provide:
- An expressivity result (a well-trained one-layer transformer can implement CoT-like GD), closely based on [1], to support the simplifying assumptions.
- In the setting of continuous regression parameter, they provide upper bounds on excess risks of (1) an ensemble aggregating random reasoning paths and on (2) the deterministic baseline. The main finding is that in high-dimensional settings, with large reasoning length, a deterministic model can overfit to label noise, whereas the sampling-based ensemble does not. The result holds for a specific way of noise injection (a linear noise transformation function), and in a very specific range of parameters.
- In the setting of binary regression parameter they lower-bound the recovery probability of the true parameter. They compare greedy decoding with majority voting. In the theoretical setup, majority voting is only proved to be better than greedy decoding for a single in-context example, where the claim is that greedy decoding can become trapped in cyclic state transitions.
- In the binary case, they fit a predictive law that forecasts accuracy from the number of sampling and thinking tokens.
- Experiments show qualitatively similar trends to what is predicted by theory.

**Compliance With Llm Reviewing Policy:**

Affirmed.

**Final Justification:**

The rebuttal has clarified my concerns about the narrative consistency of the paper: the connection of the theoretical results to the experiments and the conclusions drawn. I also appreciate the authors' comments on the generality of their theoretical assumptions. Overall, I believe that a score of 4 is appropriate.

**Key Questions For Authors:**

Questions

1. On page 7, the bold claim ‘greedy decoding can become trapped in cyclic state transitions’ is stated, but the only justification is the bounds of Theorem 6.3. Could the authors elaborate on how the bounds show this conclusion? Is this a hypothesis?
2. Section 7.2 ‘Insights for LLM inference’ presents essentially an inference-time scaling law for the discrete case. This seems like a very interesting contribution that could deserve more space. Could similar results be derived for the continuous case? How does this scaling law relate to inference scaling laws [2]?

Recommendations
1. The theoretical results use multiple variables, defined at separate places, and it’s difficult to find their definitions. E.g. $\sigma_\varepsilon$ appears in equation (5.1), but does not seem to be defined beforehand. I recommend the authors create a table (in the appendix) describing what each variable means.

Typos
- line 218 it should be ‘analyse’.
- line 218, the subsentence ‘in this section, we specific sampling algorithms …’ seems to lack a verb.
- line 220, ‘We consider same average’, the word ‘the’ is missing
- line 226-227, it should be ‘methods’
- line 262-263 (right block), it should be ‘validated’
- line 275 ‘taking … as well-trained transformer start point’ lacks ‘a’


References

[1] Huang, J., Wang, Z., and Lee, J. D. Transformers learn to implement multi-step gradient descent with chain of thought. ICLR, 2025. URL https://openreview.net/forum?id=r3DF5sOo5B.

[2] Wu, Y., Sun, Z., Li, S., Welleck, S., and Yang, Y. Inference Scaling Laws: An Empirical Analysis of Compute-Optimal Inference for Problem-Solving with Language Models, October 2024.

**Limitations:**

yes

**Strengths And Weaknesses:**

Strengths
- **Relevance and Novelty**: The paper studies an interesting and relevant problem of transformer inference. Using the CoT gradient-descent view of transformer inference to theoretically study the performance of inference algorithms is interesting and novel.
- **Soundness**: The theoretical analysis is rigorous. I didn’t check the proof in detail, but everything looks well-reasoned and clean.
- **Presentation: paper structure**: The structure of the paper is clear.

Weaknesses
- **Novelty of the theoretical framework**: The main contribution is stated to be the theoretical framework, but most of the setup is taken from [1] (which is stated). The only added contribution to the framework is the noise injection, which in itself is a simple extension. However, this limitation in novelty only concerns the paper's contribution to the theoretical framework, not the general problem formulation and theoretical analysis.
- **Significance of the results**: while the theoretical analysis is very rigorous, the results do not seem very significant. The main theorem (for continuous parameter) only shows the benefits of averaging for very specific parameter ranges and noise transformation function. It is unclear how limiting these assumptions/ranges are. In the discrete, the desired conclusions (majority voting outperforming greedy decoding) could only be established under the assumption of a single in-context example. Therefore, it seems that the predictions of the proposed theoretical framework only agree with experimental findings under very specific assumptions, limiting the impact of the theory.
- **Presentation: clarity**: While overall structured well, some parts of the paper are hard to follow. For example, in Section 2, common transformer simplifications are presented, but the reader is not told which ones the paper will decide to adapt. Furthermore, the experimental section is very condensed, and it becomes difficult to understand the relationship to the theoretical results, especially to the proposed predictive laws.

Overall, while I appreciate the paper’s rigor, I am not convinced by the strength and usefulness of the results, and am hence slightly leaning towards rejection.

---

> ### Author Rebuttal · Authors · 2026-03-31
>
> We thank the reviewer for recognizing the rigor of our analysis. The reviewer's main concern focuses on the novelty of our framework and the utility of our analysis. we emphasize that **the core components of our framework, including our noise transformation functions and sampling algorithms, are independent of specific architectural or parameter settings.** Furthermore, **our analysis generalizes to both relaxed theoretical settings and practical LLM applications.** This practical utility is validated in Sec 7.2, and we have provided additional comparisons between our method and existing inference scaling laws [2] in **R5**. Below are our detailed responses:
>
> > W1: Incremental Novelty of the Framework.
>
> **R1**: While we adapt the in-context coefficient prediction task from [1] to understand transformer test-time computing , **the core components of our framework remain independent of specific architectural or parameter settings**. This includes formulating diverse sampling techniques for both continuous and discrete settings. Rather than mere noise injection, our framework utilizes **carefully designed noise transformation functions and sampling algorithms** to replicate the stochasticity of practical LLM decoding while maintaining analytical tractability across diverse configurations. Our framework takes an initial but significant step toward bridging the gap between theory and practical language model decoding.
>
> > W2: Generalizability and Significance of Results
>
> **R2** While we utilize specific settings for analytical tractability, **our analysis can be extended to more general cases on both relaxed theoretical settings and practical LLM usage**. Our theoretical insights are validated by experiments on real-world LLMs, demonstrating the potential practical utility of our theory.
>
> For the continuous case, our assumptions are highly realistic for modern LLMs; for instance, the large hidden dimension d naturally supports the required reasoning lengths, moreover, our results hold regardless of specific eigenvalue patterns. Regarding the discrete case, our theoretical insights are validated by experiments on real-world LLMs (Sec 7.2), which go beyond single in-context examples.
>
> > W3: Sec 2 and Experiments part are hard to follow.
>
> **R3**:  _For Sec 2_: While our subsequent case studies utilizing transformer with linear attention, the simplifications introduced in Sec 2 serve as modular building blocks. Our framework remains architecture-agnostic; these foundational components can be combined to analyze test-time computation across diverse Transformer variants.
>
> _For Experimental Linkage:_ Due to space constraints, specific experimental details were moved to Appendix B. However, the relationship between Alg 4 and our theoretical analysis is directly derived from our established bounds: $O(e^{-N Δ_T^2 / 2})$ for overall accuracy and $O(e^{-μ T})$ for the probability gap $Δ_T$. We kindly refer you to our response R5 for a more detailed theoretical and empirical comparison between our method and existing inference scaling laws [2].  We will add these details in our revised version.
>
> > Q1: Technical Justification for "Cyclic State Traps"
>
> **R4**:**"Cyclic State Traps" is NOT our hypothesis, it is rigorously derived from our theoretical analysis.** As shown in Lemma F.6, we find that with high probability, the greedy decoding algorithm will only choose between two options {w', w''}, neither of which is the target w*. In this case, regardless of how many tokens are generated, the final answer will never reach the ground truth w*, resulting in poor performance for greedy decoding.
>
> > Q2: Extension to Continuous Case and Relation to Scaling Laws [2]
>
> **R5**: **Similar scaling results can be derived for the continuous setting using the bounds in Theorems 5.1 and 5.2**. While the functional form may differ since we optimize recovery loss $|w - w^*|^2$ rather than a discrete recovery rate, the dependence on N and T remains analytically tractable.
>
> **Our method provides more granular performance predictions** by jointly modeling reasoning length T and sampling number N. Unlike existing laws [2] that focus solely on N, our Alg 4 leverages the theoretical synergy between T and N to achieve superior extrapolation. As shown below, our approach is more token-efficient, achieving lower error than the simple exponential(SE)  baseline [2] with a reduced compute budget:
>
> |Benchmark|Method|Utilized Tokens($10^5$)|Error|
> |-|-|-|-|
> |GSM8K|SE|47.3|0.0089|
> ||Ours|26.0|0.0064|
> |MATH|SE|14.94|0.0447|
> ||Ours|7.63| 0.0103|
>
> > Recommendation: Variable Table
>
> **R6**: We appreciate this suggestion; $σ_ε$ represents the variance of the label noise. We will add a table in the appendix describing the meaning of each variable in our revised version.
>
> [1]Transformers learn to implement multi-step GD with CoT
>
> [2]Inference Scaling Laws: An Empirical Analysis of Compute-Optimal Inference for Problem-Solving with Language Models

---

> > ### Author Rebuttal · Reviewer_162g · 2026-04-02
> >
> > Thank you for the rebuttal! I appreciate the clarification of the cyclic state traps and comparison to existing scaling laws. I am happy to increase my score to (4), but ask the authors to incorporate these explanations to the paper to improve the accessibility of the theoretical part, and further clarify the relationship to the experiments.

---

> > > ### Author Response · Authors · 2026-04-02
> > >
> > > We sincerely thank you for your continued engagement with our work, your constructive feedback, and for raising your score.
> > >
> > > We will strictly follow your suggestions in the final version, incorporating our comparisons with existing inference scaling laws, and the explicit linkage between our theoretical bounds and experimental findings.
> > >
> > > To further clarify the relationship between our theoretical analysis and the experiments, we will move the derivation of Algorithm 4 from Appendix B.2 into the main body to further enhance readability and clarify the connection between our theory and experimental results. Specifically, we will highlight that the theoretically derived decay terms for reasoning length $T$ and sampling number $N$, namely $\mathcal{O}(e^{-\mu T})$ and $\mathcal{O}(e^{-\Delta_T^2 N/2})$, serve as the mathematical backbone for our predictive equations. Guided by these theoretical bounds, we formulate the overall accuracy as:
> > >
> > > $$\mathtt{Acc}(T,N) \approx \alpha_{(T,N)} - \beta_{(T,N)} e^{-\Delta_T^2 N/2}$$
> > >
> > > where the probability gap $\Delta_T$ is fundamentally governed by $T$ via $\Delta_T \approx \gamma - \kappa e^{-\mu T}$.
> > >
> > > Algorithm 4 then bridges theory and practice by using low-compute empirical data to fit these theoretically grounded equations (i.e., estimating parameters like $\alpha, \beta, \gamma$, and $\kappa$). This allows us to successfully predict high-compute LLM performance on real-world datasets like GSM8K and MATH. This empirical validation demonstrates that the underlying synergy between $T$ and $N$ revealed by our theoretical framework remains highly robust in modern LLMs.
> > >
> > > We are happy to addressing any further questions you may have, and thank you again for your valuable support of our work!

---

### Official Review · Reviewer_RB45 · 2026-03-13

**Soundness:** 3
**Presentation:** 3
**Significance:** 3
**Originality:** 3
**Overall Recommendation:** 4
**Confidence:** 3

**Summary:**

This paper addresses a critical gap in the theoretical understanding of transformer test-time computation. While empirical work has demonstrated that scaling test-time compute (via longer reasoning chains or sampling multiple candidates) substantially improves LLM performance, existing theoretical analyses predominantly rely on deterministic decoding frameworks that ignore the stochastic nature of practical inference.
The authors propose a novel theoretical framework that incorporates randomness directly into the transformer decoding process through noise injection and sampling mechanisms. They instantiate this framework on the canonical in-context linear regression task, analyzing both continuous and binary coefficient settings. The key contributions include: (1) a unified framework connecting stochastic decoding to test-time compute analysis; (2) theoretical proofs demonstrating when and why sampling-based methods (ensembling/majority voting) outperform deterministic approaches (greedy decoding/constant reasoning), specifically through mitigating overfitting and escaping local optima; and (3) validation through numerical simulations and practical experiments on GSM8K and MATH datasets, including a "Low-Cost-to-High Prediction" algorithm that uses theoretical insights to predict high-compute performance from low-compute observations.
Overall, the study's central aspect concerns bridging the divide between deterministic theoretical models and stochastic practical inference, providing rigorous justification for the effectiveness of test-time scaling techniques.

**Compliance With Llm Reviewing Policy:**

Affirmed.

**Key Questions For Authors:**

**Key Questions For Authors**
- **Architectural Generalization**: Your theoretical results are derived for one-layer linear attention transformers implementing gradient descent. Do you anticipate that the key insights regarding Linear NFT mitigating overfitting and majority voting escaping local optima would hold for deeper architectures with Softmax attention? If so, what additional technical challenges would arise in extending the proofs?
- **Choice of Noise Transformation**: In Section 4.1, you analyze two specific NFTs (Constant and Linear). Is there a principled way to determine the optimal NFT for a given task, or is the Linear NFT primarily motivated by its analytical tractability? Could adaptive noise schemes (where $\sigma$ varies with $t$) provide additional benefits?
- **Computational Trade-offs**: The paper establishes that sampling-based methods (e.g., majority voting) can achieve better accuracy than greedy decoding given sufficient samples $N$. However, in practice, there is a computational budget constraint. Could you provide explicit theoretical guidance on the optimal allocation between reasoning length $T$ and sample count $N$ under a fixed compute budget (e.g., total FLOPs or latency constraints)?
- **Societal Impact and Limitations**: While you mention limitations briefly in Section 8, the paper could benefit from a more explicit discussion of potential negative societal impacts. For instance, could the ability to predict high-compute performance from low-cost evaluations (Algorithm 4) be misused to optimize for benchmark performance without genuine capability improvements?

**Limitations:**

The authors acknowledge the framework's limitation to simplified transformer architectures and specific tasks (linear regression). However, they should expand the discussion of limitations, including:
- The assumption of well-trained transformers implementing exact gradient descent (Proposition 4.3), which may not capture the behavior of practically trained LLMs with suboptimal weights
- The restriction to specific noise distributions (Gaussian for continuous, categorical for binary) whereas real LLMs sample from discrete vocabularies with complex dependencies
- Potential societal impacts of test-time compute scaling, such as increased energy consumption or accessibility disparities between those who can afford extensive inference-time computation versus those who cannot

**Strengths And Weaknesses:**

**Strengths**
- **Technical Soundness**: The theoretical framework is rigorously constructed, with detailed proofs provided for excess risk bounds (Theorem 5.2) and recovery probabilities (Theorem 6.2, 6.3). The analysis correctly identifies how Linear NFT mitigates overfitting compared to deterministic baselines, supported by both theoretical derivations and numerical validations.
- **Novel Theoretical Contribution**: The paper makes a significant conceptual advance by explicitly incorporating stochastic sampling into the transformer decoding analysis, bridging the gap between deterministic theoretical models and practical LLM inference. The distinction between Constant NFT and Linear NFT provides new insights into why certain sampling strategies work better than others.
- **Clear Organization and Presentation**: The paper is well-structured, with intuitive motivation in the Introduction followed by formal framework development. The parallel treatment of continuous (Section 4.1) and binary (Section 4.2) coefficient cases effectively demonstrates the framework's versatility. Figure 1 successfully illustrates the alignment between theoretical predictions and real-world LLM behavior.
- **Practical Relevance and Validation**: Beyond theoretical analysis, the authors validate findings on GSM8K and MATH datasets and propose a practical "Low-Cost-to-High Prediction" algorithm (Algorithm 4). This demonstrates the framework's potential utility for real-world inference optimization, not just abstract theory.
- **Timely Research Direction**: As test-time compute scaling becomes increasingly critical for modern LLMs (e.g., OpenAI o1, DeepSeek-R1), this work addresses an urgent need for theoretical understanding of inference-time techniques, providing foundational insights for future algorithmic development.

**Weaknesses**
- **Architectural Simplifications**: The analysis relies on one-layer linear attention transformers implementing gradient descent (Proposition 4.3), which, while standard in theoretical ICL literature, may not fully capture the behavior of deep, practical LLMs with softmax attention and complex layer interactions. The extent to which results generalize to deeper architectures remains unclear.
- **Limited Task Scope**: The framework is instantiated only on in-context linear regression tasks. While this provides analytical tractability, it remains uncertain whether the insights regarding noise injection and majority voting directly transfer to more complex reasoning tasks (e.g., mathematical reasoning, code generation) where the solution space structure differs significantly.
- **Insufficient Discussion on Computational Trade-offs**: Although the paper analyzes the benefits of increasing sample number $N$ and reasoning length $T$, it lacks explicit theoretical guidance on optimal resource allocation under fixed computational budgets (e.g., FLOPs or latency constraints). In practice, practitioners must balance these factors, and clearer theoretical trade-offs would enhance practical utility.
Underexplored Societal Implications: While the paper briefly mentions limitations in Section 8, it insufficiently addresses potential negative societal impacts of extensive test-time computation, such as increased energy consumption, carbon footprint, and accessibility disparities between well-resourced and constrained deployment environments.
- **Assumption of Ideal Training**: The analysis assumes well-trained transformers that implement exact gradient descent. In practice, trained LLMs may exhibit suboptimal weights, training instabilities, or distribution shifts that deviate from this idealized behavior, potentially limiting the framework's predictive accuracy for real-world models.

---

> ### Author Rebuttal · Authors · 2026-03-31
>
> We sincerely appreciate your thorough review of our work. We value your constructive feedback, and we have addressed your questions, concerns, and identified limitations in detail below:
>
> > [W1&Q1]: Architectural Simplifications & Generalization
>
> **R1**: **Yes, our theoretical framework and main results will naturally generalize to deeper/softmax attention**. Specifically, even in these complex settings, deterministic methods tend to overfit and lack sufficient capacity to explore the solution space, causing them to underperform sampling-based test-time computation approaches as reasoning length and sample count increase.
>
> The primary technical challenge in extending our proofs is the varying complexity of the prediction update ($Δ w_t = w_{t+1} - w_t$). Unlike linear attention, where $Δ w_t$ is simply the vanilla gradient, softmax and multi-layer attention introduce complex non-linear dynamics. Recent studies demonstrate that these complex updates can be approximated as debiased gradient descent [1] and multi-step gradient descent [2], respectively. Thus, by modeling $Δ w_t$ as a single gradient-based optimization step, our analysis can effectively extend to these deeper architectures. We leave the rigorous proof of this extension for future work.
>
> [1]In-Context Linear Regression Demystified:Training Dynamics and Mechanistic Interpretability of Multi-Head Softmax Attention
>
> [2]Transformers learn to implement preconditioned gradient descent for in-context learning
>
> > [W2&W4&L1&L2]:Limited Task Scope and Assumption of Ideal Training
>
> **R2**: The premise that transformers can implement exact gradient descent is supported by both theoretical and empirical evidence [3], serving as a solid starting point to _theoretically_ analyze test-time computation behavior.
>
> Beyond rigorous theoretical analysis. **We further demonstrate the practical application of our theoretical framework with experiments on practical LLMs**, we develop an algorithm (in Sec 7.2) based on our analysis that successfully predicts real-world LLM performance under computationally expensive settings. **Moreover, to bridge the gap between idealized theory and real-world models, our algorithm introduces parameterized modifications to account for actual task and model complexity.** As illustrated in Fig 3, this algorithm uses empirical data from configurations with relatively low computational budgets (low reasoning tokens T and sampling numbers N) to fit free parameters specifically to the target model, successfully predicts real-world LLM performance under computationally expensive settings, providing evidence that our framework effectively characterizes actual model behaviors despite the initial theoretical simplifications.
>
> [3] Transformers learn to implement multi-step gradient descent with chain of thought
>
> > [W3&Q3]:Discussion on Computational Trade-offs
>
> **R3**: Our primary contribution is establishing a foundational theoretical framework for understanding transformer test-time computing behavior. Consequently, our analysis focuses on case studies demonstrating the framework's broad applicability. While deriving strict computational trade-offs is an important downstream application, it falls outside our immediate scope and remains a promising direction for future work.
>
> Our current framework can already provide insights into downstream applications like computational trade-offs. Specifically, by utilizing Alg 4 to approximate the performance Acc(N,T), practitioners can systematically identify the optimal reasoning settings (T,N) under a fixed compute budget (N × T). Such downstream applications are highly relevant and warrant further exploration in future studies.
>
> > [Q2]:Choice of Noise Transformation
>
> **R4**: We chose the Linear/Constant NFT because it is a widely adopted noise injection method that maintains analytical tractability [4]. Furthermore, adaptive noise schemes could indeed provide additional benefits. By treating the noise variance σ as a learnable parameter, we can optimize the noise injection function through gradient descent. This approach enables the sampling algorithm to dynamically achieve an optimal balance between exploration and exploitation during inference.
>
> [4]Benign overfitting in linear regression
>
> > Societal Impact and Limitations
>
> **R5**: Thank you for pointing this out. While the main aim of this work is to develop a theoretical framework to understand test-time computation techniques, we recognize the broader implications of test-time scaling. Our method has the potential to help identify the optimal computational trade-off to achieve strong model performance, which can guide the efficient utilization of computing resources and aid in reducing energy consumption, whether operating under extensive or limited inference-time budgets. We appreciate your insight and will explicitly incorporate these vital discussions into a broadened Limitations and Broader Impacts section in our final revision.

---

> > ### Author Rebuttal · Reviewer_RB45 · 2026-04-04
> >
> > The authors have provided convincing responses to all my technical concerns. The justification for architectural simplifications (with clear extension paths to deeper models), validation on real-world LLMs via Algorithm 4, and commitments regarding societal impact are all satisfactory.

---

### Decision · Program_Chairs · 2026-04-30

**Decision:**

Accept (regular)

**Comment:**

Reviewers praised the paper's rigorous theory on stochastic test-time computation and its practical validation predicting real-world LLM performance. There were some initial concerns raised by the reviewers, but after the rebuttal period all are advocating for acceptance.